# Reduced methane emissions in transgenic rice genotypes are associated with altered rhizosphere microbial hydrogen cycling

Ling-Dong Shi[1,18], Maria Florencia Ercoli [2,18], Junhyeong Kim[3], Artur Teixeira de Araujo Junior[2], Katerina Estera-Molina[1,4], Subah Soni [2], Tracy Satomi Weitz[2], Alexandra M. Shigenaga[2], Ilija Dukovski[5,6,7], Rohan Sachdeva [1], Halbay Turumtay[8,9,10], Katherine B. Louie [11], Benjamin P. Bowen [9,11], Suzanne M. Kosina [9], Henrik V. Scheller [8,9,12], Jennifer Pett-Ridge [1,4,13], Daniel Segrè [5,6,14,15,16], Trent R. Northen [9,11], Pamela C. Ronald [1,2,8] ✉ & Jillian F. Banfield [1,3,12,17] ✉

Rice paddies significantly contribute to atmospheric methane ($CH_4$). Here, we show that two independent rice genotypes overexpressing genes for *PLANT PEPTIDES CONTAINING SULFATED TYROSINE* (*PSY*) reduce cumulative $CH_4$ emissions by 38% (PSY1) and 58% (PSY2) over 70 days of growth compared with controls. Genome-resolved metatranscriptomic data from PSY rhizosphere soils reveal lower ratios of gene activities for (mostly hydrogenotrophic) $CH_4$ production versus consumption, decreased activity of $H_2$-producing genes, and increased activity of bacterial $H_2$ oxidation pathways. Metabolic modeling using metagenomic and metabolomic data predicts elevated $H_2$ oxidation and suppressed $H_2$ production in the PSY rhizosphere. Assembled genomes of rhizosphere $H_2$-oxidizing bacteria are enriched in genes utilizing gluconeogenic acids compared with $H_2$-producing counterparts, and their activities are likely stimulated by elevated levels of gluconeogenic acids, primarily amino acids, in PSY root exudates. Overall, our study indicates that decreased $CH_4$ emissions are due to a lower amount of $H_2$ available for hydrogenotrophic methanogenesis and provides a powerful strategy to mitigate $CH_4$ emissions from increasingly widespread rice cultivation.

Methane ($CH_4$) is a greenhouse gas with a 27–30 times greater warming effect than carbon dioxide ($CO_2$) over a 100-year period. Atmospheric $CH_4$ has recently increased to a record-high concentration of 1933 ppb (May, 2025)[1]. One of the most important $CH_4$ sources is the waterlogged, anaerobic soils of rice paddies, which support the growth of methanogenic archaea that catalyze the final step in anaerobic decomposition of organic matter, generating $CH_4$[2,3]. Although produced $CH_4$ can be partially oxidized in situ through aerobic and/or anaerobic methane oxidation[4,5], the imbalance between $CH_4$

production and oxidation leads to rice paddies contributing 7-17% of global $CH_4$ emissions[6,7]. Emissions are predicted to further increase due to the expansion and intensification of rice cultivation needed to feed the fast-growing human population[8].

Numerous approaches have been explored to mitigate $CH_4$ emissions from rice paddies, primarily focusing on farming practices such as managing organic matter, water, and nitrogen[9]. Genetic optimization of rice plants, for example, allocating less photosynthate to roots, is an alternative $CH_4$ mitigation strategy. Transgenic

introduction of a single transcription factor gene, barley *SUSIBA2*, into rice plants confers more nutrients to aboveground biomass and less to roots, therefore reducing $CH_4$ formation[10]. Similarly, a loss-of-function allele of rice *gs3* (a gene on chromosome 3 controlling grain size) allocates more carbon to the grain and less to the roots and causes lower production of $CH_4$[11].

Another approach to reduce $CH_4$ emissions is to alter the rice root phenotype. Through programmed cell death, rice roots develop aerenchyma, which facilitates gas diffusion and enables efficient oxygen supply from the aerial parts of the plant to the submerged roots and surrounding soil. Simultaneously, plants deposit suberin and lignin to form complete apoplastic barriers in the outer root layers to reduce radial oxygen loss[12,13]. It has been reported that rice genotypes with a higher capacity for internal oxygen diffusion grow larger root systems in waterlogged soils and are associated with lower $CH_4$ emissions due to their stronger ability to oxygenate the soil[14,15]. Previously, we and others have shown that small tyrosine-sulfated peptides, such as the PLANT PEPTIDES CONTAINING SULFATED TYROSINE (PSY) family, play central roles in coordinating root development. Acting as extracellular signals, these peptide hormones synchronize cellular activities across tissues by binding to leucine-rich repeat receptor-like kinases (LRR-RLKs), thereby triggering downstream signaling cascades[16]. A critical post-translational modification, sulfation of a tyrosine residue by TYROSYLPROTEIN SULFOTRANSFERASE (TPST), is essential for receptor recognition and signaling efficiency[17-20]. In *Arabidopsis thaliana*, overexpression of PSY genes enhances root growth by promoting cell elongation and increasing the size of mature cortical cells[21-23]. These peptides exert their effects through three known LRR-RLK receptors, PSYR1, PSYR2, and PSYR3, recently identified as direct mediators of PSY signaling[22,24]. Notably, PSY1 signaling in *Arabidopsis* activates genes involved in secondary metabolism that regulate the extent of cell elongation and transition to maturation. Some of these metabolites are subsequently secreted into the rhizosphere as part of the root exudate profile[21,23,25].

Given that methanogenic archaea inhabit the rice root rhizosphere and that substrates for $CH_4$ production (e.g., $H_2/CO_2$ and acetate) ultimately derive from root exudates[26], we hypothesized that modulation of the rice root phenotype would affect microbiome activity and $CH_4$ cycling. To this end, we generated rice plants with altered seminal roots by overexpressing the rice *PSY* genes (*OsPSY1* and *OsPSY2*) and cultivated them in a greenhouse, along with the control Kitaake genotype, in soils from rice paddies. Plant physiological traits, root exudate chemistry, $CH_4$ emissions, rhizosphere microbial community composition, and in situ gene activities were compared over 70 days of rice growth.

## Results

### Generation of rice plants overexpressing rice *PSY1* and *PSY2* genes

To identify putative rice PSY genes, we surveyed the Nipponbare and Kitaake genomes using bioinformatic approaches as previously described[25,27], and identified nine candidate PSY genes (Supplementary Fig. 1a). Sequence alignment of the mature peptide domain revealed high conservation between Arabidopsis (AtPSYs) and rice (OsPSYs) peptides (Supplementary Fig. 1b), suggesting that OsPSY peptides may regulate rice root growth similarly to AtPSY peptides[21-23,25,28]. Supporting this hypothesis, we demonstrated that Kitaake seedlings, grown on solid media or hydroponically, developed longer seminal roots when treated with synthetic OsPSY1 (Fig. 1a, b). Publicly available transcriptomic datasets[29,30] reveal that *OsPSY1*, also known as *CHALKY GRAIN 5* (*OsCG5*)[31], is predominantly expressed in spikelets, with lower expression levels in other rice tissues, including roots (Supplementary Fig. 1c–e). We hypothesize that the root growth effect observed after synthetic peptide treatment may result from the presence of leucine-rich repeat receptor-like kinases (LRR-RLKs) in root tissues that

recognize OsPSY1. It has been recently shown in Arabidopsis, that PSY peptides directly bind to LRR-RLKs from subfamily XI to trigger a signaling cascade that regulates root development[22]. The conservation of LRR-RLKs across plant species suggests that functionally similar receptors likely exist in rice and may mediate comparable developmental responses. In contrast to *OsPSY1*, other members of this gene family, such as *OsPSY2*, displayed more ubiquitous expression across different rice organs (Supplementary Fig. 1d). Specifically, *OsPSY2* expression in rice roots increased progressively in the differentiation zone (DZ), a pattern previously noted for *AtPSY1* promoter activity[23] (Supplementary Fig. 1e).

To further investigate the roles of *OsPSY1* and *OsPSY2* in rice development, we independently overexpressed each gene in the Kitaake genetic background. We generated >50 independently transformed lines using the constitutive UBIQUITIN (Ubi) promoter to drive the expression of *OsPSY1* and *OsPSY2* (*Ubi::OsPSY1* and *Ubi::OsPSY2*). We self-pollinated the resulting transgenic plants and conducted analyses on the progeny derived from four independently transgenic lines, including *Ubi::OsPSY1#7*, *Ubi::OsPSY1#9* (hereafter referred to as PSY1-9), *Ubi::OsPSY1#17* (PSY1-17), *Ubi::OsPSY1#32*, *Ubi::OsPSY2#8* (PSY2-8), *Ubi::OsPSY2#19*, *Ubi::OsPSY2#21* and *Ubi::OsPSY2#34* (PSY2-34).

Expression analysis of root tissues showed that progeny derived from the *Ubi::OsPSY1* and *Ubi::OsPSY2* transgenic lines accumulated significantly higher levels of *OsPSY1* or *OsPSY2* transcripts compared with Kitaake (Fig. 1c, d). Consistent with the synthetic peptide treatment analysis, rice plants expressing higher levels of *OsPSY1* or *OsPSY2* exhibited longer seminal roots (Fig. 1e, f) and also carried the transgene (Supplementary Fig. 1f, g).

Given that in Arabidopsis, *AtPSY1* regulates root growth by controlling the extent to which cells elongate before reaching their final differentiated size[22,23], we measured the length of mature cortical cells in the DZ of PSY1-9 and PSY1-17 genotypes. Consistent with the phenotypes observed in Arabidopsis, we found that the roots of rice genotypes accumulating higher levels of *OsPSY1* exhibited longer mature cortical cells (Fig. 1g, h). Additionally, the amount of lignin in PSY1-17 and PSY2-34 plants was significantly lower compared with Kitaake (Fig. 1i). A similar phenotype was observed in the roots of Arabidopsis plants with higher levels of *AtPSY1*, where lignin deposition in the tracheary elements of the protoxylem occurred further away from the root tip compared with the wild type[23]. Collectively, these results suggest that PSY peptides from different species may regulate root development through similar molecular mechanisms.

Finally, we assessed aerenchyma formation in a selected subset of the rice PSY1 (PSY1-9 and PSY1-17) and PSY2 (PSY2-8 and PSY2-34) genotypes. We found that in all the transgenic lines tested, the percentage of aerenchyma formation was significantly higher compared with Kitaake (Fig. 1j–l). However, no differences in aerenchyma were observed between the PSY1 and PSY2 lines (Fig. 1j–l). Given the observed phenotypic modifications in roots of rice seedlings, particularly the longer seminal roots and enhanced aerenchyma formation, we hypothesized that PSY genotypes may impact the rhizosphere microbial community and associated $CH_4$ cycling.

### Growth of PSY rice genotypes in agronomic soils mitigated $CH_4$ emissions compared with controls

To assess the effect of PSY overexpression on $CH_4$ emissions, we cultivated three independent transgenic rice lines—PSY1-9, PSY1-17, and PSY2-34—along with the control Kitaake plants in a greenhouse using soils collected from a rice paddy in Davis, CA. We also included bulk soil controls that were not planted. Throughout the experiments, all rice lines developed similarly to Kitaake, with heading observed at 55 days after being transferred to the greenhouse (Fig. 2a). By 70 days post-transplant, both PSY1 and PSY2 genotypes were slightly shorter than Kitaake (Fig. 2b and Supplementary Fig. 2a), but they produced

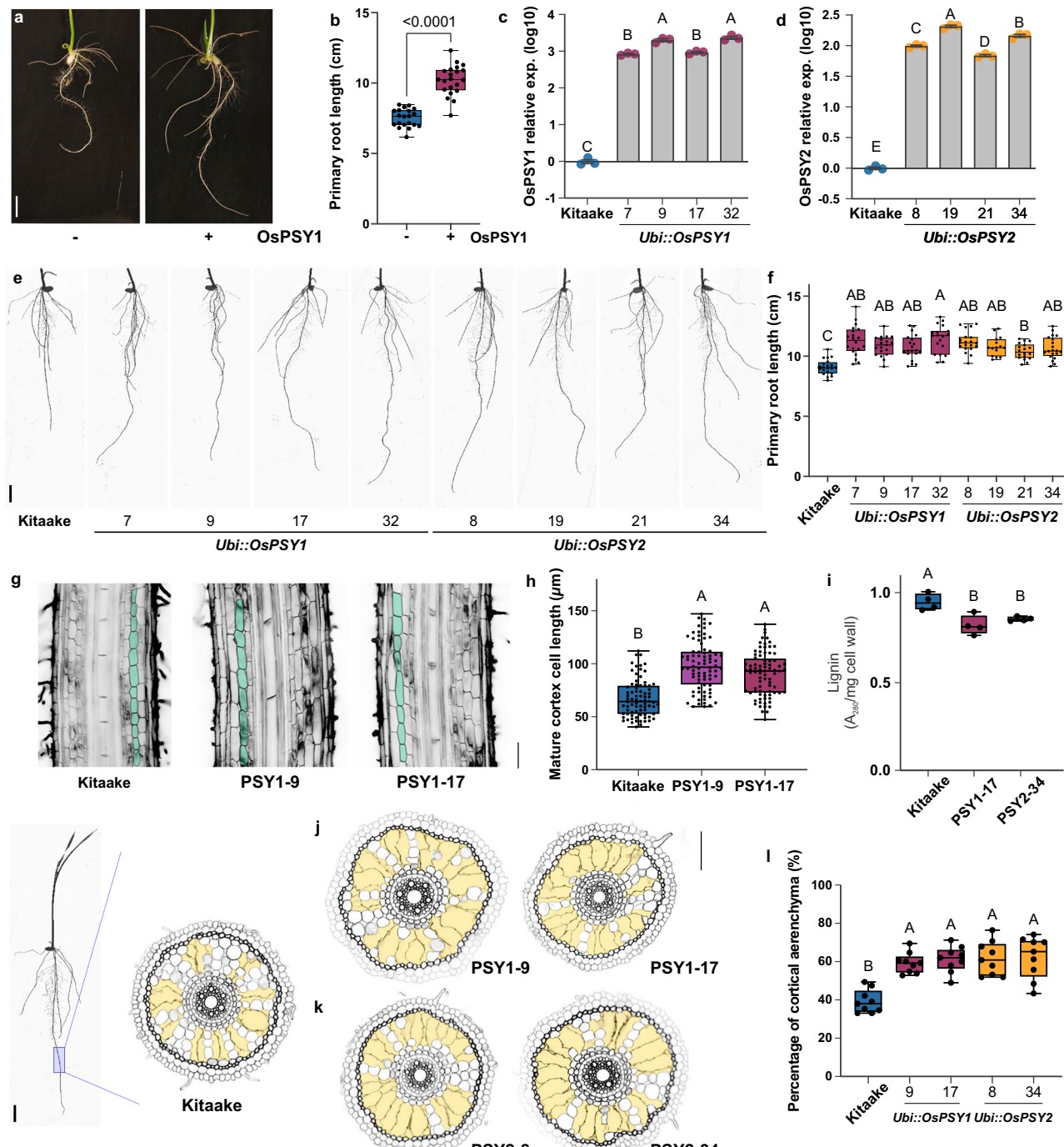

**Fig. 1 | Overexpression of OsPSY1 and OsPSY2 enhanced primary root growth in rice. a** Root phenotype and **b** root growth in Kitaake seedlings grown hydroponically for 7 days on 1× MS with (n = 21 seedlings) or without 500 nM of synthetic OsPSY1 (n = 20 seedlings). Expression of **c** *OsPSY1* or **d** *OsPSY2* in four independent homozygous transgenic lines derived from parental lines *Ubi::OsPSY1* (#7, #9, #17, and #32) and *Ubi::OsPSY2* (#8, #19, #21, and #34), compared with the Kitaake control. Expression was measured by RT-qPCR in three biological replicates and normalized to the mean value obtained in control plants. Bars represent mean values; error bars indicate the standard error of the mean (SEM). **e** Root phenotype and **f** root growth in seedlings of Kitaake (n = 20), and independent homozygous transgenic lines *Ubi::OsPSY1* #7 (n = 20), #9 (n = 19), #17 (n = 20), and #32 (n = 19), and *Ubi::OsPSY2* #8 (n = 20), #19 (n = 15), #21 (n = 18), and #34 (n = 20), grown on 1× MS vertical plates. **g, h** Quantification of cortical cell length (highlighted in green) in the differentiation zone of Kitaake and *Ubi::OsPSY1* (#9 and #17) seedlings grown for 7 days on 1× MS vertical plates (n = 4 seedlings, 20 cells per seedling). **i** lignin content (n = 4 replicates of 15 complete seedlings root system). **j–l** Depictions of the percentage of cross-sectional area occupied by aerenchyma (highlighted in yellow) at 2.5–3.5 cm from the root tip in Kitaake, *Ubi::OsPSY1* (#9 and #17) (**j**) and *Ubi::- OsPSY2* (#8 and #34) (**k**) independent homozygous transgenic seedlings grown for 7 days on 1× MS vertical plates (n = 9 seedlings). On the left, the blue shaded box in the rice seedling indicates the region from which root tissue was collected for imaging (scale bar is 1 cm). **b, f, h, i, l** The data are shown as box and whisker plots combined with scatter plots; each dot indicates the measurement of the designated parameter listed on the y axis of the plot. The box spans the interquartile range (25th–75th percentiles), the center line denotes the median, and whiskers extend to the smallest and largest observed values. **b** P values are calculated by a two-tailed Student's t test. **f, h, i, l** Different letters indicate significant differences, as determined by one-way ANOVA followed by Tukey's multiple comparison test (P < 0.05). Scale bars in (**a, e**) are 1 cm. Scales bar in (**g, j, k**) are 100 μm.

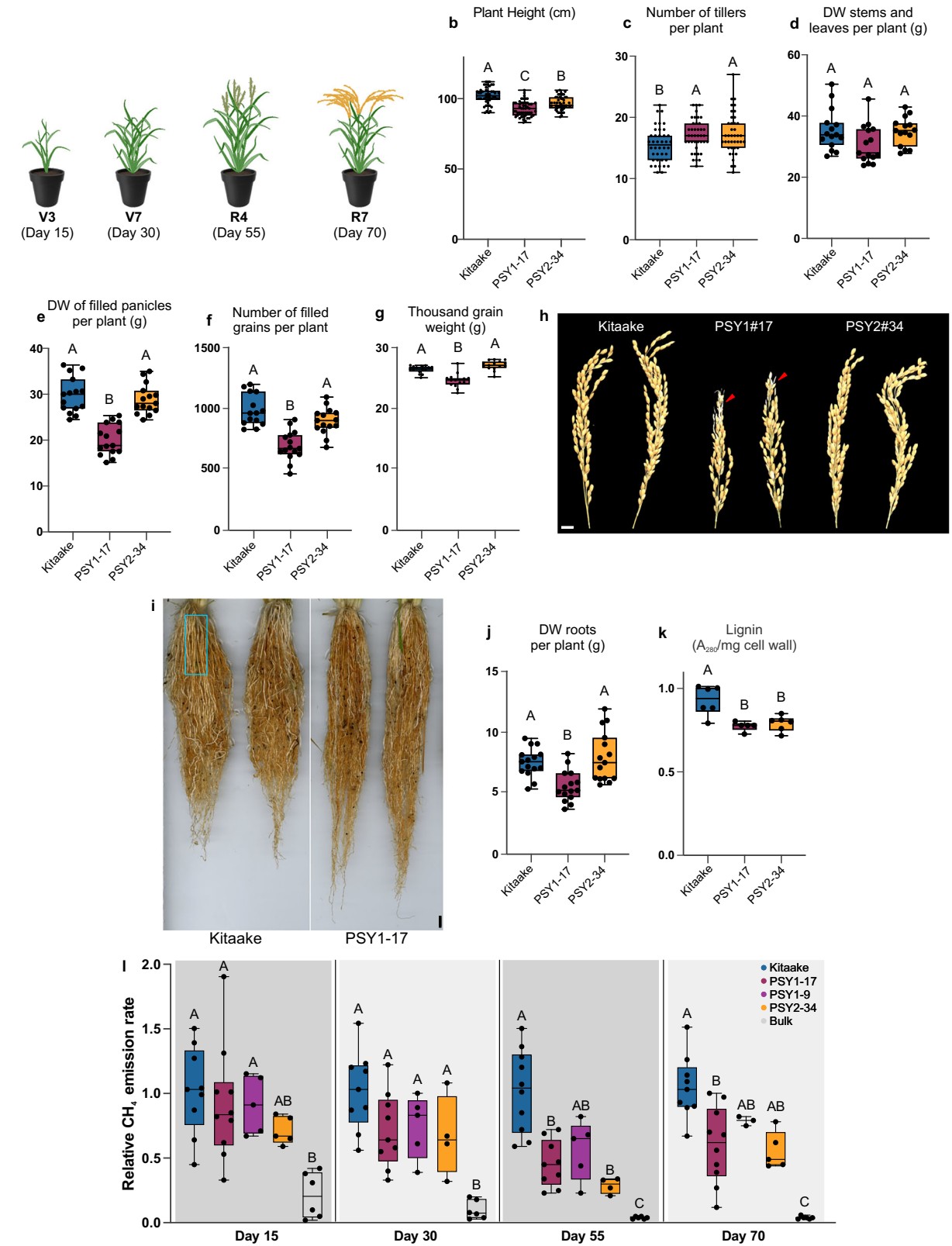

significantly more tillers (Fig. 2c and Supplementary Fig. 2b), resulting in comparable dry weights of stems and leaves per plant at the end of the plant life cycle (Fig. 2d). PSY1 genotypes showed partially filled panicles, fewer seeds, and lower thousand-grain weight (TGW) per plant, compared with Kitaake (Fig. 2e–g and Supplementary Fig. 2c). These phenotypes could be attributed to abnormal panicle development, as we found that the tips of the panicles in PSY1 genotypes often contained smaller and unfilled seeds, a phenotype not seen in PSY2 plants (Fig. 2h). Consistently, no significant differences in seed production occurred in the PSY2 genotypes compared with Kitaake (Fig. 2e–h).

We also investigated whether the longer seminal root phenotype observed in young seedlings persisted in older PSY plants. We found that PSY1-17 plants developed longer roots 30 days after transplanting,

**Fig. 2 | Kitaake plants with elevated expression of *OsPSY1* or *OsPSY2* exhibited reduced CH$_4$ emissions. a** Illustrations depicting the plant developmental stage at each time point selected for methane measurements, coupled with rhizosphere sample collection in greenhouse experiments. Created in BioRender. Shigenaga, A. (2026) https://BioRender.com/ aw5xl5s. Phenotypic profiling of Kitaake, PSY1-17, and PSY2-34 including **b** plant height (cm), **c** number of tillers per plant, **d** dry weight (DW) from stems and leaves per plant (g), **e** DW of filled panicles per plant (g), **f** number of filled grain per plant, **g** thousand-grain weight (g), **h** representative panicles images (scale bar = 1 cm). **i** Representative image of the root system phenotype from Kitaake and PSY1-17 rice plants 30 days after transplanting. The light blue square in the Kitaake picture highlights the region of the root system where rhizosphere and lignin samples were collected (scale bar = 1 cm). **j** DW from complete root systems per plant (g), and **k** root lignin content. Phenotypic measurements were taken 70 days after transplanting for (**b**) (*n* = 50 plants), (**c**) (*n* = 40

plants), and (**k**) (*n* = 6 Kitaake and PSY2-34 plants, *n* = 5 PSY1-17 plants), while (**d**–**f**, **g**, **j**) were analyzed at the end of the plant life cycle 100 days after transplanting (*n* = 15 plants). **l** Methane emission rates from different rice genotypes and bulk soil controls are normalized to the mean value obtained in Kitaake plants. The number of biological replicates (*n*) for Kitaake, PSY1-17, PSY1-9, PSY2-34, and bulk soil, respectively, were: day 15 (*n* = 9, 10, 5, 5, 6), day 30 (*n* = 9, 9, 5, 4, 6), day 55 (*n* = 10, 9, 5, 4, 6), and day 70 (*n* = 9, 10, 3, 5, 6). The data shown in (**b**–**f**, **g**, **j**, **k**, **l**) are box and whisker plots combined with scatter plots; each dot indicates the measurement of the designated parameter listed on the *y* axis of the plot. The box spans the interquartile range (25th–75th percentiles), the center line denotes the median, and whiskers extend to the smallest and largest observed values. Different letters indicate significant differences, as determined by one-way ANOVA followed by Tukey's multiple comparison test (*P* < 0.05).

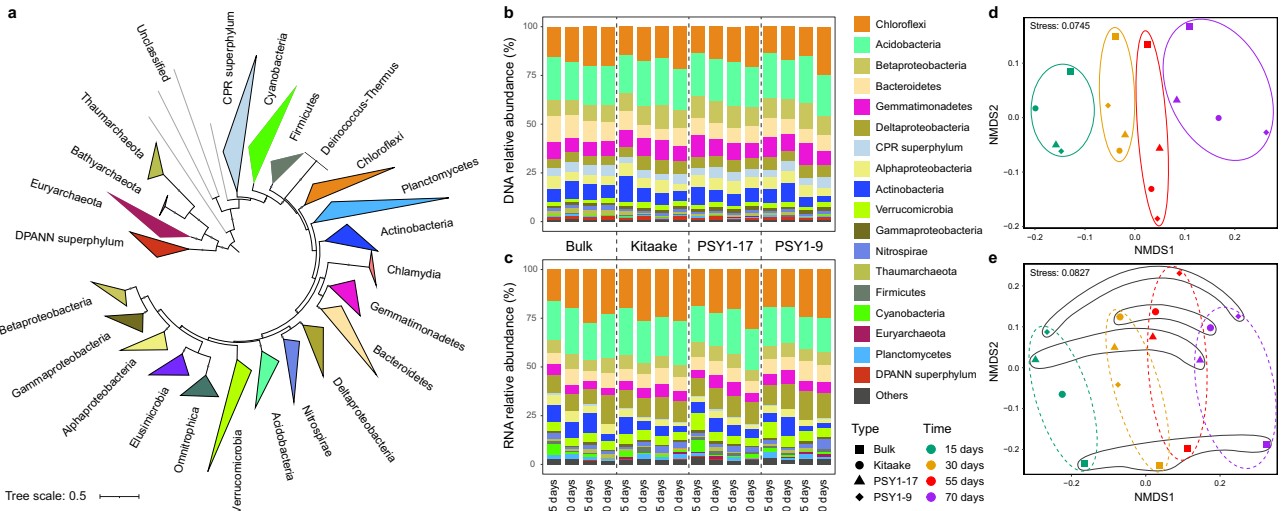

**Fig. 3 | Microbial community structure and activity. a** Microbial community membership based on phylogeny of identified rpS3 amino acid sequences. Black dots indicate support values ≥ 80% calculated based on 1000 replicates. Relative abundance (**b**) and in situ activity (**c**) of microbial communities at the phylum level, measured at four timepoints in the bulk soil and three rice genotypes. "Others" indicates the summed values for rare lineages. Non-metric multidimensional scaling (NMDS) ordination of microbial community structure (**d**) and transcriptional

abundance (**e**) with symbol meanings shown in "Type" and "Time" keys. Dissimilarity scores (*R*) were calculated by analysis of similarity (ANOSIM) test: for community structure (**d**), *R* = 0.539, *P* = 0.001 between cultivation time (highlighted by ovals), and *R* = 0.0729, *P* = 0.238 between rice genotypes; for community activity (**e**), *R* = 0.265, *P* = 0.006 between rice genotypes (highlighted by solid ovals), and *R* = 0.195, *P* = 0.020 between cultivation time (highlighted by dashed ovals).

---

although their root dry biomass was significantly lower than that of Kitaake plants (Fig. 2i and Supplementary Fig. 2d, e). Similar differences in root dry biomass were observed at the end of the plant life cycle (Fig. 2j). Additionally, lignin content in the roots of adult plants revealed that, consistent with findings in young seedlings, PSY1-17 plants accumulated less lignin than Kitaake (Fig. 2k). Although PSY2 plants also had reduced lignin content, their root biomass remained comparable to that of Kitaake (Fig. 2j, k).

Despite these phenotypic differences, no significant differences in soil CH$_4$ emissions were observed between the PSY1-9, PSY1-17, and PSY2-34 genotypes (Fig. 2l). For these analyses, CH$_4$ emissions were measured in situ using a Picarro gas analyzer, and soil samples were collected for microbiological analyses throughout the rice growth period (i.e., 15, 30, 55, and 70 days after transplanting). Negligible CH$_4$ emissions were observed from the bulk soil controls, while differentiated emissions were recorded from pots with PSY and Kitaake rice plants over the 10-week growth period (Fig. 2l and Supplementary Fig. 3a). All PSY rice genotypes emitted substantially less CH$_4$ compared with Kitaake, with reductions of up to ~30% observed after heading (55 and 70 days; Fig. 2l and Supplementary Fig. 3a). Notably, cumulative CH$_4$ emissions throughout the growth period for PSY1 and PSY2 genotypes showed an average decrease of ~38% and ~58% relative to Kitaake, respectively (Supplementary Fig. 3b). Given the crucial role

of microorganisms in the production and consumption of CH$_4$, we hypothesized that distinct microbial communities might accumulate in the rhizospheres of the different rice genotypes, potentially affecting the rice CH$_4$ emissions. Because both PSY1 and PSY2 genotypes displayed comparable reductions in CH$_4$ emissions, we focused on PSY1 genotypes for further study.

### Rice PSY genotypes affected microbial activity but not microbial community structure

To explore the difference in root microbiomes, we extracted and sequenced total DNA and RNA from 72 soil samples (3 replicates for each of the bulk soils and 5 replicates for each planted with the PSY1 rice genotypes and Kitaake control, at each collection time point) (Supplementary Data 1). We used assembled ribosomal protein S3 (rpS3) sequences to perform an initial census of microbial diversity, including organism type and abundance[32], in 16 representative samples obtained by combining replicates. In total, 4,087 rpS3 sequences were identified and grouped into 1,656 non-redundant, approximately species-level clusters. Phylogenetic analysis based on rpS3 sequences classified all detected microorganisms into at least 4 archaeal and 14 bacterial phyla (Fig. 3a). By mapping metagenomic and metatranscriptomic reads to rpS3 genes, the most abundant and active organisms were Proteobacteria, Chloroflexi, Acidobacteria, Bacteroidetes,

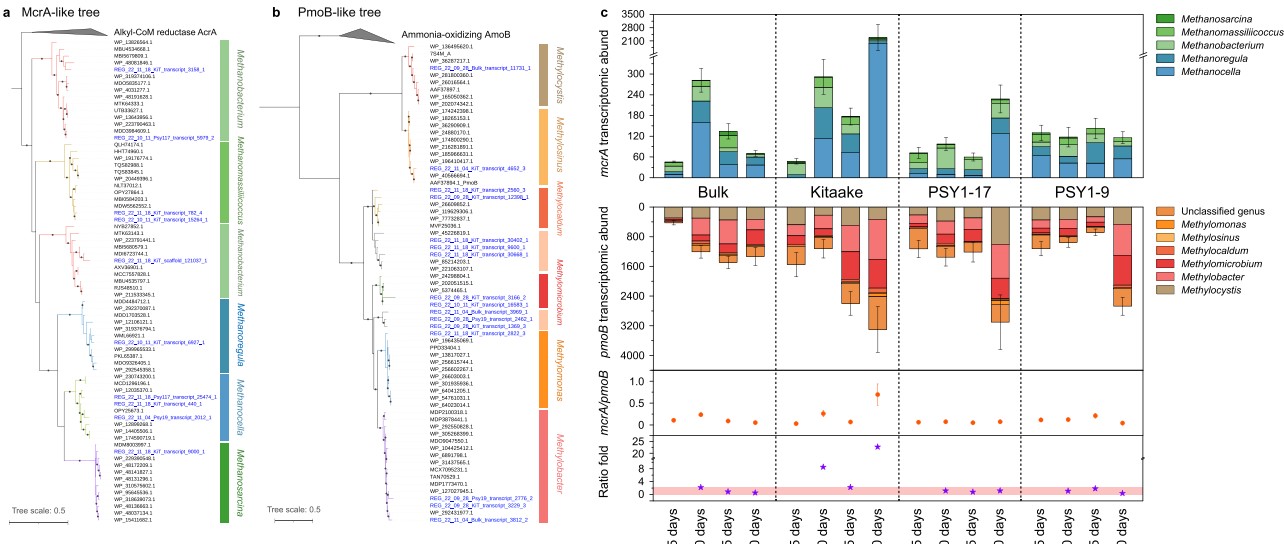

**Fig. 4 | Taxonomic affiliation and in situ activity of genes involved in CH₄ production and oxidation. a**, **b** Phylogeny of identified McrA and PmoB sequences. Black dots indicate support values ≥ 80% calculated based on 1000 replicates. Blue clades are proteins identified in the study. **c** Transcriptional activity and taxonomic affiliation of *mcrA* and *pmoB* genes in the bulk soil and rhizosphere soils of different rice genotypes at four timepoints during the cultivation period. Error

bars indicate standard deviations of sum values in replicates ($n = 3$ for bulk soil and $n = 5$ for rice genotypes). Ratios of *mcrA* to *pmoB* were calculated by dividing *mcrA* transcriptomic abundances by *pmoB*. Ratio change folds in each group were calculated using ratios at the first sampling point (15 days after transplanting) as baselines. The pink bar indicates the range between 0 and 2.

Gemmatimonadetes, and Actinobacteria (Fig. 3b, c), consistent with previous soil surveys[33–35]. Between 1 and 5% of sequences were assigned to the "Candidate Phyla Radiation" (CPR) bacteria and DPANN archaea (Diapherotrites, Parvarchaeota, Aenigmarchaeota, Nanoarchaeota and Nanohaloarchaeota), both of which were recently reported in paddy soils[36,37].

Non-metric multidimensional scaling analyses (NMDS) show clear differences in community structure and microbial activity between soils collected from the rhizosphere and unplanted bulk soil (Fig. 3d, e), indicating a strong effect from the rice plants[38]. For the planted groups, cultivation time had a large and statistically significant effect on the community structure (dissimilarity score of up to 0.539), whereas rice genotype did not (Fig. 3d). Notably, however, microbial activity was significantly influenced by both time and genotype with comparable contributions (dissimilarity scores of 0.265 and 0.195, respectively; Fig. 3e).

### Lower activity of CH₄-producing genes in microbiomes associated with PSY1 rice genotypes

CH₄ fluxes from rice paddy soils reflect the combined impact of the activities of methanogens and methanotrophs. A search for methylcoenzyme M reductases (McrA) that participate in methanogenesis and anaerobic CH₄ oxidation identified multiple homologs assigned to *Methanobacterium*, *Methanomassiliicoccus*, *Methanoregula*, *Methanocella*, and *Methanosarcina* (Fig. 4a). None of the recovered McrA sequences are similar to those found in anaerobic CH₄-oxidizing archaea, suggesting that anaerobic methanotrophs were either absent or too rare to be detected. Many particulate methane monooxygenases (PmoB) genes that enable aerobic CH₄ oxidation were identified and taxonomically assigned to several bacterial genera (Fig. 4b). The soluble form of methane monooxygenase was not detected. These genes are typically only expressed under copperlimiting conditions[39] (e.g., <0.1 μM versus 0.4 μM in our fertilized water; Supplementary Data 2).

In general, the transcriptional activity of *mcrA* increased with cultivation time in the three rice genotypes. For example, the activity of *Methanocella mcrA* in the Kitaake genotype increased by approximately one order of magnitude in the last growth stage (70 days,

Fig. 4c). The transcriptional activity of *pmoB* also increased with cultivation time, and we observed more active terminal oxidases in microbes associated with the PSY1 genotypes late in the experiment (Supplementary Fig. 4), suggestive of higher O₂ concentrations. However, the ratio of *mcrA* to *pmoB* remained stable over the whole growth period, except for Kitaake (Fig. 4c). Given the importance of the balance between activities that produce and consume CH₄, we calculated the fold changes of *mcrA*/*pmoB* ratios for each group relative to that at the corresponding first sampling point (i.e., 15 days after transplanting), and observed notable increases in Kitaake but negligible changes in PSY1 genotypes (Fig. 4c). This result indicates that, over the rice growth period, *mcrA* activity was promoted much more than *pmoB* in Kitaake, leading to a higher ratio of CH₄ production to consumption, as compared to the PSY1 genotypes.

### PSY1 rice genotypes supported higher activities of microbial H₂-consuming genes but lower activities of H₂-producing genes

Expressed *mcrA* transcripts were predominantly from *Methanocella*, *Methanoregula*, and *Methanobacterium* (>90%; Fig. 4c). Archaea of these genera all use H₂, instead of acetate or methylated compounds, to produce CH₄[40]. Thus, we identified genes involved in H₂ production and consumption and tracked their expression levels as a function of genotype and time. Two types of hydrogenases that catalyze the reversible conversion between H₂ and protons/electrons[41], namely [NiFe] and [FeFe] hydrogenases, were identified and divided into four and three subtypes, respectively, using protein phylogeny (Fig. 5a, b). The most active clade was [NiFe] Group 1 hydrogenases that oxidize H₂ (Fig. 5c). The transcriptional activities of this group were generally higher in the two PSY1 genotypes than in the Kitaake control, particularly in the first and last growth stages where the differences were significant (15 and 70 days; Fig. 5d). The [NiFe] Group 4 and [FeFe] Group A1 (further classified by HydDB[42]) are predicted to produce H₂[43]. Transcript quantification shows [NiFe] Group 4 hydrogenases were more active than [FeFe] Group A1 hydrogenases, and the combined activity of these two H₂-producing hydrogenases was significantly higher in Kitaake compared with the two PSY1 genotypes, especially in the second and last growth stages (30 and 70 days; Fig. 5d). Based on the transcriptomic profile analysis, we hypothesize that the

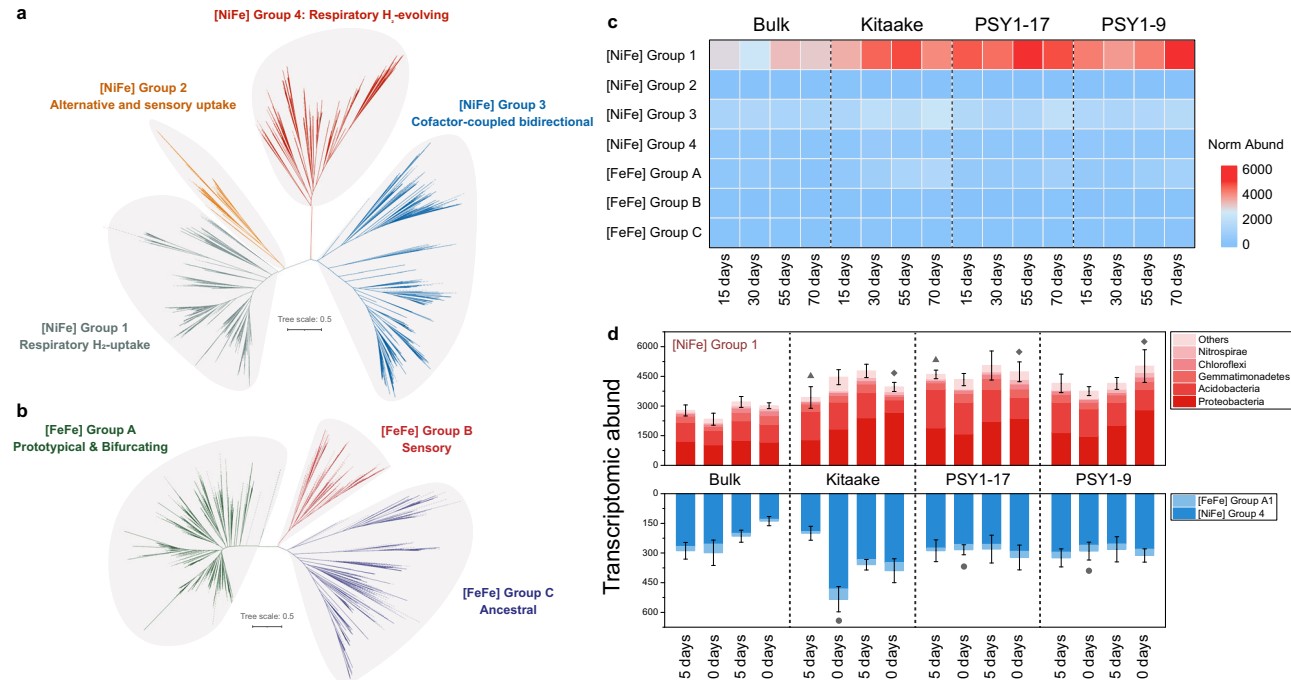

**Fig. 5 | Classification and activity of hydrogenases.** Phylogeny of identified [NiFe] hydrogenases (**a**) and [FeFe] hydrogenases (**b**). Black dots indicate support values ≥ 80% calculated based on 1000 replicates. Dashed branches are references. **c** Transcriptional activity of classified hydrogenase groups in different rice types over the cultivation period. **d** Transcriptomic abundances of [NiFe] Group 1 hydrogenases resolved by organisms of origin, and the combination of [FeFe] Group A1 and [NiFe] Group 4. Error bars indicate standard deviations of sum values in replicates ($n = 3$ for bulk soil and $n = 5$ for rice genotypes). Symbols (i.e., circles,

triangles, diamonds) indicate significant differences between wild type (Kitaake) and PSY genotypes (PSY1-17 and PSY1-9) at the same cultivation time, tested by the Wilcoxon Rank Sum Test. For [NiFe] Group 1 hydrogenases in Kitaake versus PSY1-17, $p = 0.030$ (two-sided) on 15 days and $p = 0.036$ (two-sided) on 70 days; in Kitaake versus PSY1-9, $p = 0.037$ (two-sided) on 70 days. For the combination of [FeFe] Group A1 and [NiFe] Group 4, $p = 0.012$ (two-sided) in Kitaake versus PSY1-17 and $p = 0.020$ (two-sided) in Kitaake versus PSY1-9 on 30 days.

microbiomes of the PSY1 genotypes might produce less and consume more $H_2$ than those in Kitaake. As a result, the pool of hydrogen available for $CH_4$ production in soils from the PSY genotypes would be expected to be lower than that of the Kitaake control.

$H_2$ may also be produced as a byproduct of $N_2$ fixation (one molecule of $H_2$ per $N_2$ fixed). The nitrogenase gene (*nifH*) was used as a marker of $N_2$ fixation activity and associated $H_2$ production[44,45]. The transcriptional activity of *nifH* generally increased with cultivation time and was linked to Delta-, Gamma-, and Alphaproteobacteria (Supplementary Fig. 5). Like $H_2$-producing hydrogenases, *nifH* activity was significantly higher in Kitaake than in the PSY1 genotypes, especially in the last two growth stages (55 and 70 days).

### Altered PSY1 root exudates modulate activities of $H_2$-metabolizing microorganisms

To explore the influence of rice phenotypes on rhizosphere microbiome activity, we performed metabolomic analysis on root tissues and exudates from axenic rice seedling roots to identify compounds secreted by the PSY1 genotypes. Metabolites were extracted using methanol and profiled on an HPLC-MS platform. Using gradient-based hydrophilic interaction liquid chromatography (HILIC) in positive-ion mode, we identified a total of 3612 features across all exudate samples, of which 2047 were shared between the Kitaake and PSY1 genotypes (Supplementary Fig. 6a). Principal component analysis (PCA) showed a clear separation between PSY1 and Kitaake exudate samples (Supplementary Fig. 6b). Among the shared features, 168 (8.3%) were significantly differentially abundant in PSY1 exudates ($|\log_2FC| > 0.5$, $p < 0.05$), including 146 novel and 15 depleted features (Supplementary Fig. 6c). A subset of these differentially abundant features was putatively identified and classified using Global Natural Products Social Molecular Networking (GNPS) (Supplementary Data 3–5). Most

of the differentially abundant metabolites in PSY1 root exudate samples belong to the same compound classes, including 'amino acids', 'alkaloids', and 'shikimates and phenylpropanoids' (Supplementary Fig. 6d). Similarly, in root tissues, we identified 2,237 shared features, with PCA indicating an even stronger genotypic separation (Supplementary Fig. 6e, f). Of these, 272 (12%) were differentially abundant in PSY1 tissues, featuring 208 new and 80 depleted metabolite signals (Supplementary Fig. 6g), again reflecting shifts in similar compound classes (Supplementary Fig. 6h). Overall, more features increased than decreased in PSY1 samples. Similar results were obtained from HILIC in negative ionization mode and reverse phase (C18) in both positive and negative ionization modes and are detailed in Supplementary Data 3–5, and Supplementary Figs. 7–9. Altogether, our analysis indicated significant changes in the abundance of compounds in the PSY1 root exudates compared with Kitaake. Approximately 90% of the metabolites enriched in the PSY1 metabolome are predicted to be involved in microbial gluconeogenic pathways (Supplementary Data 6). We also noted the presence of two putative flavonoids, iso-rhamnetin 3-rutinoside and malvidin-3-glucoside, that differentially accumulated in exudate samples from the PSY1 genotypes but were absent from Kitaake, a feature observed in previous studies in Arabidopsis[23].

To assess whether these metabolic differences reflected changes in the total amount of carbon released, we measured the concentration of non-purgeable organic carbon (NPOC) in the root exudate samples. No significant differences in total carbon content were detected between Kitaake and PSY genotypes, indicating that PSY over-expression primarily alters the composition, instead of the overall quantity, of secreted metabolites (Supplementary Fig. 10).

To determine whether the altered exudate composition of PSY genotypes can directly influence microbial activity, we conducted soil

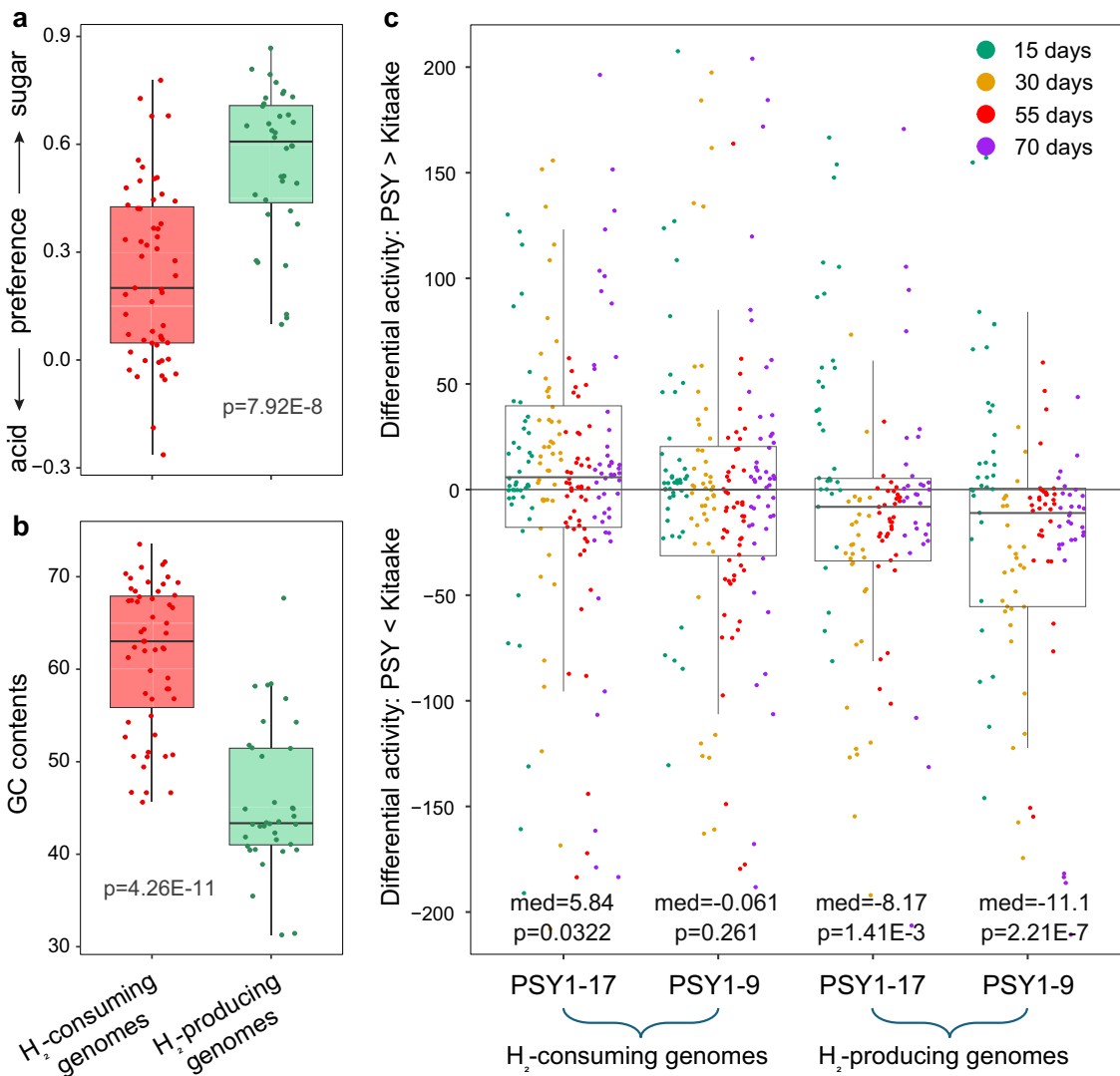

**Fig. 6 | Metabolic preference and activity of H₂-consuming and H₂-producing genomes.** Predicted sugar-acid-preference values (**a**) and GC contents (**b**) of H₂-cycling populations. Colored dots indicate genomes in two groups ($n = 55$ and $n = 36$). Box plots show lower and upper quartiles, and median values in each genome group. Whiskers indicate maximum and minimum values excluding outliers. Statistical differences were calculated using the Wilcoxon rank sum test.

**c** Differential activities of genomes involved in H₂ cycling. Differential activities were calculated by subtracting overall transcriptional activities in PSY1 rice genotypes from those in Kitaake. Colored dots indicate genomes in different rice growth stages. Box plots show the same information as in (**a**, **b**). Median differential activities were statistically compared to zero using a one-sample Wilcoxon Signed Rank Test.

incubation assays using paddy soil from the same rice field used in our greenhouse experiments. Exudates collected from PSY1- and PSY2-overexpressing rice seedlings were added as amendments, with exudates from Kitaake and sterile water as controls. CH₄ emissions were then monitored over time. No CH₄ flux was detected in water-treated soils, confirming that exudates were necessary to stimulate methanogenesis. Soils treated with Kitaake exudates exhibited consistently higher CH₄ emissions than those treated with PSY1 or PSY2 exudates, mirroring trends observed in the greenhouse-grown plant experiments (Supplementary Fig. 11).

Next, we analysed all the microbial genomes reconstructed from the rhizosphere ($n = 296$) and found 17 had transcriptional activities that were significantly upregulated in soils of rice PSY genotypes relative to the Kitaake control (log2FoldChange $\geq 1$, $p \leq 0.05$; Supplementary Fig. 12). The genomes of five of the 17 encode H₂-consuming hydrogenases, including those affiliated with Pyrinomonadaceae, Burkholderiaceae, Haliangiaceae, Rhodocyclaceae, and Dissulfurispiraceae, but none contain H₂-producing enzymes (Supplementary Data 7). This result suggests that H₂-consuming bacteria may

respond to the PSY rice genotype-derived exudates differently than organisms that produce H₂. We therefore surveyed the entire H₂-consuming ($n = 55$) and H₂-producing ($n = 36$) populations and calculated glycolytic sugar versus gluconeogenic acid preference values (SAP) using the method described previously[46]. We found that the H₂-consuming organisms have much higher preferences for gluconeogenic acids than the H₂-producing organisms, and their genomes have higher GC contents, likely reflecting this metabolic preference[46] (Fig. 6a, b; Supplementary Data 7). Consequently, H₂-consuming organisms had higher transcriptional activities in association with the PSY1 genotypes, whereas their H₂-producing counterparts were more active in the Kitaake rhizosphere (Fig. 6c).

To further investigate how rice physiological traits affect microbial H₂ cycling, we built metabolic models using representative, high-quality genomes (completeness >90%, contamination <5%). We then predicted H₂ fluxes in response to variations in PSY1-secreted exudate compounds and O₂ concentrations inferred from aerenchyma densities of the different rice genotypes (Supplementary Data 8). The only identified compounds under-secreted by PSY1 compared with Kitaake

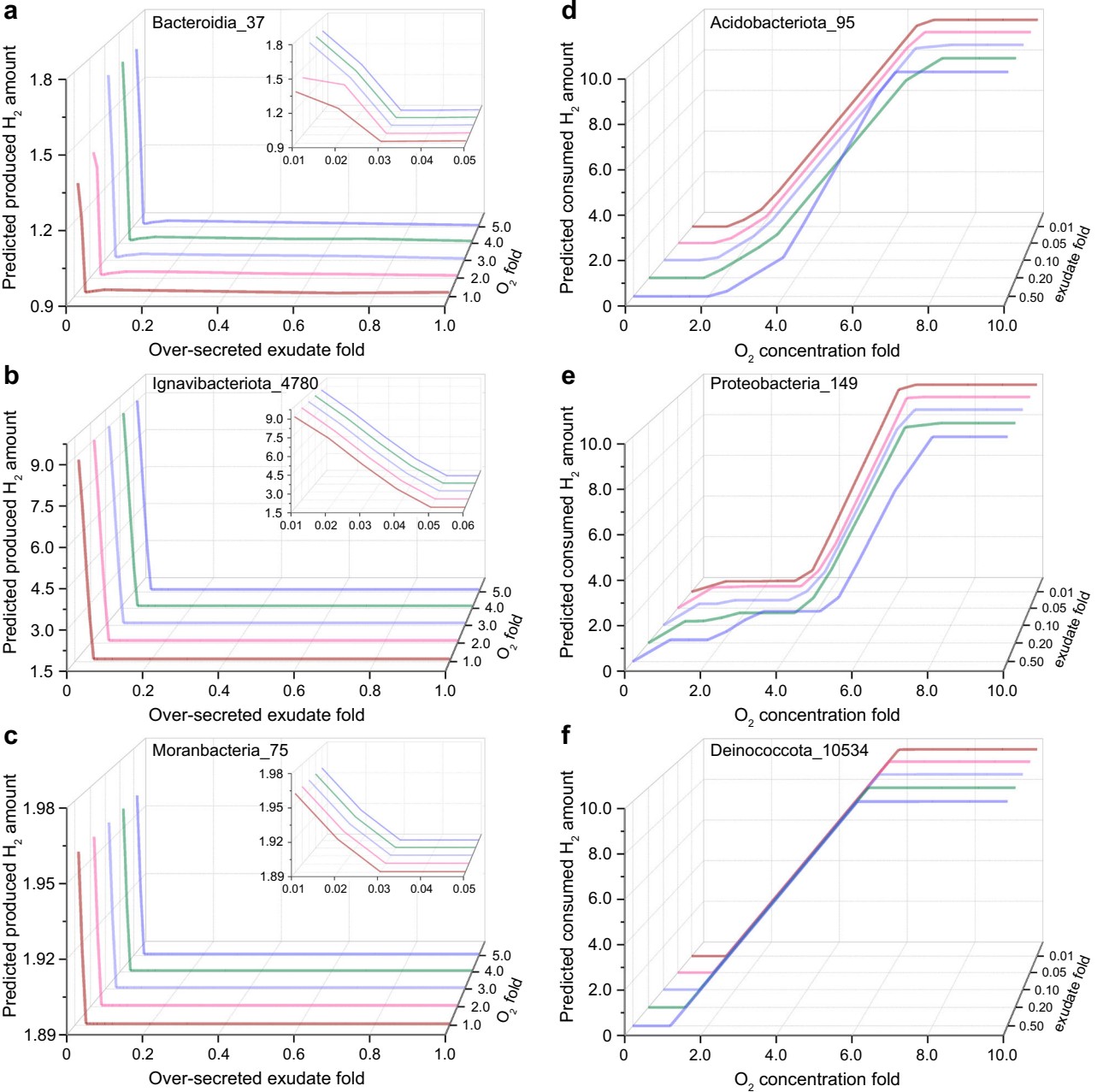

**Fig. 7 | Predicted H₂ flux by metabolic modeling using representative high-quality genomes.** Inferred H₂ production (**a**–**c**) and H₂ consumption (**d**–**f**) under varying PSY over-secreted exudate and O₂ concentrations. Fold changes of over-secreted exudates and O₂ are defined when compared to essential nutrients that do not differ across rice genotypes. Insets in **a**–**c** show details of H₂ production at low over-secreted exudate concentrations. H₂ consumption in **d**–**f** was calculated by subtracting final H₂ concentrations from initial. Genome name abbreviations:

Bacteroidia_37, REG_22_10_11_KiT_metabat2_37 in Supplementary Data 7; Ignavibacteriota_4780, REG_22_11_18_KiT_metabat2_4780; Moranbacteria_75, REG_22_11_18_KiT_metabat2_75; Acidobacteriota_95, REG_22_09_28_Psy117_metabat2_95; Proteobacteria_149, REG_22_10_11_Psy117_concoct_149; Deinococcota_10534, REG_22_09_28_KiT_vamb_10534. All detailed information of these genomes is present in Supplementary Data 7.

were two glycerolipids, putatively 1-oleoyl-2,3-di-linoleoyl-sn-glycerol and 1,2-di-hexadecanoyl-3-(9Z,12Z,15Z-octadecatrienoyl)-sn-glycerol, and one dipeptide containing leucine (Supplementary Data 3). However, none of the modeled H₂-related organisms can metabolize these compounds. As a result, the PSY1 under-secreted metabolites are predicted to have no effect on H₂ fluxes. In contrast, when over-secreted compounds increase from 0.01 to 0.05-fold, all three modeled H₂-producing bacteria from different phyla (Bacteroidia, Ignavibacteria, and Moranbacteria) show the same trend, with maximal produced H₂ decreasing (Fig. 7a–c). Beyond this range, H₂ production is predicted to plateau. Microbial growth increased with increasing

concentration of over-secreted compounds and plateaued when the H₂ production plateaued, except for Bacteroidia_37, which continually increased its biomass over the whole simulation (Supplementary Fig. 13a–c). The results suggest that the modeled H₂-producing bacteria can use over-secreted PSY1 compounds to support growth, but not via pathways that generate H₂. Increased O₂ availability has little or no effect on H₂ production (Fig. 7a–c).

The H₂-consuming organisms from Acidobacteriota, Proteobacteria, and Deinococcota are modeled to oxidize more H₂ (and accordingly yield more biomass) in the presence of higher O₂ concentrations that are associated with higher aerenchyma formation in

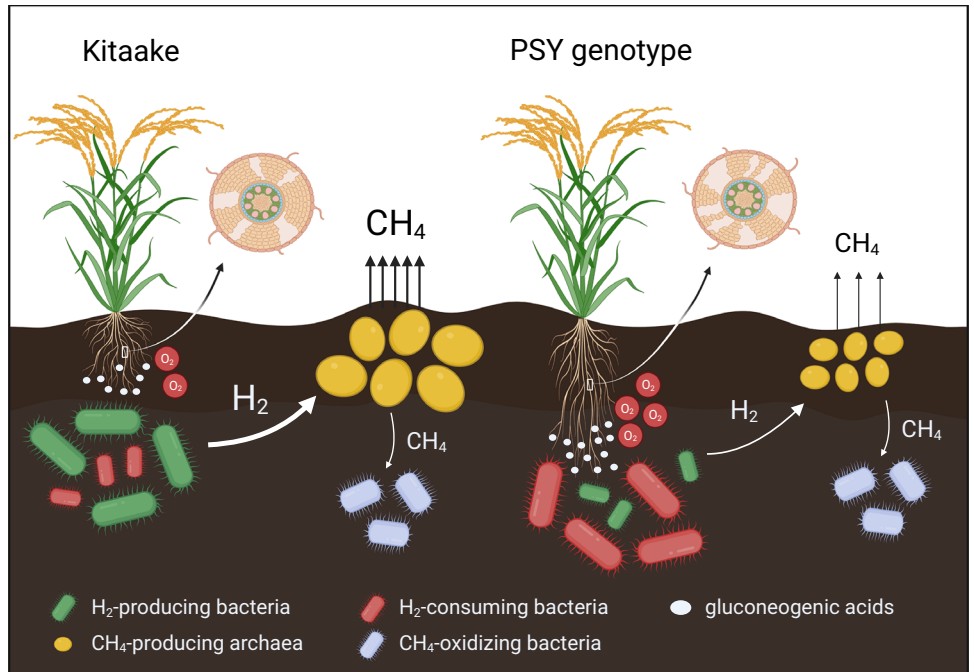

**Fig. 8 | Proposed mechanism for the role of PSY rice genotypes in mitigating CH₄ emissions.** PSY rice genotypes have longer roots with more aerenchyma compared with Kitaake. $O_2$ is transported from the atmosphere to the rhizosphere through aerenchyma. PSY rice roots secrete more gluconeogenic acids (white circles), mostly amino acids, that can facilitate the activities of $H_2$-consuming bacteria (indicated by cell sizes), thus reducing the total $H_2$ pool for $H_2$-dependent methanogenesis. The produced $CH_4$ was partially oxidized in situ by methanotrophic bacteria, and the rest was emitted into the atmosphere. Created in BioRender. Shi, L. (2026) https://BioRender.com/gq0748f.

PSY1 rice genotypes (Fig. 7d–f and Supplementary Fig. 13d–f). The PSY1 over-secreted exudate compounds have a negligible influence on $H_2$ oxidation, yet still contribute substantially to the accumulation of biomass of $H_2$-consuming organisms (Supplementary Fig. 13d, e). These modeling results support the metagenomics-based hypothesis that PSY1 rice genotypes enhance the growth and activity of $H_2$-consuming organisms due to their distinct aerenchyma characteristics and exudate profiles.

## Discussion

Rice is the most important staple food for half the world's human population. It is estimated that the global rice production will continue to increase from 506 million tonnes in 2020 to 567 million tonnes by 2030[47]. During cultivation, rice paddies emit millions of tons of $CH_4$ annually and contribute -10.1% of global agricultural greenhouse gas emissions[48]. Here, we tested whether overexpression of the rice genes, *OsPSY1* or *OsPSY2*, could affect root growth, exudate composition, activities of rhizosphere microbiomes and $CH_4$ emissions, potentially providing a route to decrease the climate impact of rice production. To our knowledge, this is the first study to use an integrated multi-omic approach to elucidate how specific rice genotypes mitigate $CH_4$ emissions. The research highlights how the intersecting activities of $H_2$-metabolizing, $CH_4$-producing and other microbes in rhizosphere communities modulate $CH_4$ cycling (Fig. 8).

The $CH_4$ production in our study was derived mainly from hydrogenotrophic methanogens (>90%; Fig. 4c), consistent with extensive prior research[11,49,50]. Previous rice field studies using isotope-labeled substrates revealed that the presence of electron acceptors such as sulfate and nitrate channelled acetate towards $CO_2$, and that $CH_4$ was mainly produced from $H_2/CO_2$-dependent methanogenesis[51,52]. Dependence on $H_2$ is likely explained by the insufficient acetate concentrations, which are insufficient for methane production. Acetoclastic methanogens (e.g., *Methanosarcina*) require 0.2–1.2 mM acetate to activate $CH_4$ formation[53,54], whereas

hydrogenotrophic methanogens (e.g., *Methanobacterium*) only need 2.8–10 Pa $H_2$[55,56]. Low acetate concentrations may be due to the rapid in situ utilization by co-existing organisms that outcompete methanogens for the acetate. One possible route for acetate consumption is syntrophic acetate oxidation (SAO) that uses the reverse Wood-Ljungdahl (WL) pathway (sometimes combined with the glycine cleavage system) and hydrogenases to produce $CO_2$ and $H_2$, the substrates for hydrogenotrophic methanogenesis[57–59]. However, none of our hydrogenase-harboring microbial genomes encode complete WL pathways, although some unclassified Firmicutes members that may include SAO bacteria were detected at low abundance (<1%; Fig. 3). An alternative way for acetate consumption is the reduction of nitrate and sulfate, observed in previous studies[51,52]. This activity is indicated by the expression of nitrate and sulfate reductases in our system (Supplementary Figs. 14 and 15). The substrates (i.e., -2.0 mM of both nitrate and sulfate) were periodically added to water used to fertilize the rice plants during growth (Supplementary Data 2). Given the absence of WL pathways in hydrogenase-containing microorganisms, the homoacetogenesis process that relies on $H_2$ oxidation to produce acetate appears to be negligible in our study.

Although for all wild-type and transgenic rice genotypes the dominant pathway for $CH_4$ production was $H_2$-based methanogenesis, PSY1 and PSY2 genotypes mitigated the cumulative $CH_4$ emissions by up to 58%, relative to the Kitaake control (Supplementary Fig. 3). The most straightforward distinction between the wild-type and transgenic genotypes is the lower expression of *mcrA* and the lower ratio of transcripts for $CH_4$ production versus oxidation (*mcrA/pmoB*) in microbiomes associated with the PSY1 genotypes (Fig. 4c). The much lower expression of *Methanocella mcrA* in the PSY1 genotypes compared to the Kitaake control (Fig. 4c) in the last growth stage is in line with the lower measured $CH_4$ emissions (Fig. 2l). This difference may be partially attributed to higher $O_2$ concentrations in the PSY rhizosphere due to more aerenchyma in the PSY compared with Kitaake roots (Fig. 1j–l). Consistent with this, the activity of terminal oxidases in

microbes associated with the PSY1 genotypes was higher than for microbes associated with the Kitaake control (Supplementary Fig. 4). Although *Methanocella* cells encode genetic machinery to resist oxidative stress and can persist in aerated environments[60], their methanogenesis activity would be inhibited by oxygen exposure until conditions become anoxic[61].

Another factor that may have affected $CH_4$ production is the availability of the methanogenesis substrate, $H_2$. In rice paddies, $H_2$ is typically produced during microbial fermentation, e.g., of plant-derived sugars, and can be consumed via diverse processes[62]. We found much higher activities of $H_2$-consuming hydrogenases but lower activities of $H_2$-producing counterparts associated with the PSY1 genotypes (Fig. 5). This likely reduced the $H_2$ available for hydrogenotrophic methanogenesis, compared to that with the Kitaake.

Activities of the root-associated $H_2$-metabolizing microorganisms can be altered by rice root characteristics. As larger root systems can be associated with increased $CH_4$ emissions in paddy fields[63], rice breeding efforts for mitigating $CH_4$ emissions have aimed to generate plants with smaller root systems[10,11]. However, overexpression of rice *PSY* genes led to longer roots (Fig. 2k) and only the PSY1 genotype displayed reduced root biomass (Fig. 2i), compared with the control. The distinct developmental phenotypes observed in rice plants overexpressing *OsPSY1* and *OsPSY2* may be driven by differences in receptor binding and tissue-specific expression. Similar behavior has been observed in other small peptide families, such as the CLE family[64]. Still, both OsPSY1 and OsPSY2 genotypes appear to share common molecular mechanisms contributing to comparable reductions in $CH_4$ emissions (Fig. 2l).

Given the similar reduction in $CH_4$ emissions despite differences in PSY1 and PSY2 root biomass, we considered whether the enrichment of PSY exudates in gluconeogenic acid metabolites (mostly amino acids) could alter rhizosphere microbial $H_2$ metabolism, with implications for $CH_4$ production (Supplementary Data 6). $H_2$-consuming microorganisms that are more invested in the use of these compounds than $H_2$-producing bacteria had significantly higher activities associated with the PSY1 genotypes, suggestive of lower hydrogenotrophic methanogenesis in the PSY1 rhizosphere (Fig. 6). Consistent with this hypothesis, soil incubations with exudates collected from PSY rice roots exhibited lower $CH_4$ emissions than those amended with exudates from Kitaake. While our soil exudate incubation experiments clearly show the effect of rice root exudates on microbial shifts and associated $CH_4$ emissions, it is unclear whether this is caused by a specific exudate compound or by a combination of multiple compounds. It is also possible that root-associated environmental changes (e.g., altered redox conditions) may affect the rhizosphere microbiome. Future research is warranted to investigate these possibilities.

Our metabolic modeling results also indicate that PSY1 over-secreted compounds suppress $H_2$ production activity (Fig. 7). Intriguingly, the modeling predicts the increase in biomass of $H_2$-producing bacteria despite the decreased $H_2$ generation (Supplementary Fig. 13). This may be explained by the high relative abundance of amino acids in PSY1 exudates (Supplementary Data 6). Root-associated $H_2$-producing bacteria may partly rely on extracellular amino acids that can be converted to pyruvate and acetyl-CoA, and thus rely less on fermentation of sugars to form $H_2$[65,66]. The rice-derived amino acids could also alleviate the necessity for microbial nitrogen fixation (Supplementary Fig. 5), reducing the formation of byproduct $H_2$[67]. Conversely, microbial $H_2$ consumption was barely affected by the PSY1 over-secreted compounds (Fig. 7). Overall, our metabolic modeling predicts that the exudate profiles of PSY1 rice genotypes may inhibit the activities of $H_2$-producing microorganisms and stimulate those of the $H_2$-consuming counterparts. This result suggests that in the PSY1 rhizosphere, the bacterial metabolic activities further diminish the pool of $H_2$ available for methanogenesis, thus mitigating overall $CH_4$ emissions (Fig. 8). Future studies could isolate and co-culture the organisms used in the models to validate the inferred microbial responses to root-derived environmental changes.

It has been demonstrated that exudation rate and composition vary throughout plant development[68]. In this study, we collected exudates from seedlings grown under axenic hydroponic conditions. This method offers several advantages: precise control over the chemical milieu, adequate recovery of fresh exudates, avoidance of mineral sorption biases, and minimization of microbial degradation[69]. However, hydroponic growth can alter exudate profiles compared with soil systems, often increasing total carbon release but reducing metabolite diversity due to the absence of rhizosphere feedbacks[70]. Given these methodological trade-offs, future studies could integrate both axenic hydroponic and soil-based exudate sampling across multiple developmental stages to gain a more comprehensive and ecologically relevant understanding of how PSY influences plant–microbe interactions and biogeochemical cycles.

While our greenhouse-based experiments using agronomic soil collected from a rice basin provide clear evidence that rice genotypes overexpressing PSY genes reduce $CH_4$ emissions, controlled environments do not fully replicate the complexity of field conditions, where factors such as temperature fluctuations and seasonal dynamics may play major roles. We consider our study a critical first step in uncovering mechanistic links between specific plant genotypes, root exudate chemistry, and $CH_4$ cycling. To validate and extend these findings to agronomically relevant conditions, field trials incorporating multi-omics profiling, similar in scope to our current approach, are necessary. Such future research will be essential to determine the generalizability and robustness of PSY-mediated reduced $CH_4$ emissions across various soil types and growing conditions.

## Methods

### Plant materials and growth conditions

*Oryza sativa* ssp. *Japonica* cultivar Kitaake was used throughout this study as the wild-type (wt) background. Rice seeds were dehulled, surface-sterilized in 20% bleach for 30 minutes, and thoroughly washed with autoclaved water. The seeds were grown in 1× Murashige and Skoog (1× MS) salt mixture with 1% sucrose (pH 5.65), either as a solid medium containing 0.3% Gellex (Gellan Gum, Caisson Laboratories) or hydroponically in flasks with 100 mL of the same media without Gellex. For solid medium conditions, seeds were sown on plates, sealed with Micropore tape for gas exchange, and placed vertically for 7 days. For hydroponic conditions, seeds were first germinated on plates with water for 2 days in darkness, then transferred to flasks for another 5 days. The incubators used for rice germination and growth were set with a 14-h-light/10-h-dark photoperiod at 28 °C/24 °C.

For synthetic peptide treatments in hydroponic conditions, peptide (or water for untreated plants) was added to the MS media before transferring the plants. The length and concentration of the peptide treatment are specified in the figure legends. The synthetic OsPSY1 peptide was obtained from Pacific Immunology (Ramona, CA, USA), is tyrosine sulfated, and was diluted in ddH2O to a final concentration of 1 mM. For each experiment, MS media was freshly prepared and cooled for 1 hour in a 55–60 °C water bath after autoclaving before adding chemicals.

For primary root elongation, seedlings grown hydroponically or in solid media were photographed 7 days after germination, and the primary root length was measured with Fiji Is Just ImageJ[71].

### Sequence alignment and peptide conservation analysis

To assess the conservation of the PSY peptides across species, we extracted the mature peptide sequences from *Arabidopsis thaliana* and *Oryza sativa* PSY family members. The peptide sequences were defined as the 13-amino acid mature domain, including the conserved aspartate-tyrosine (DY) motif, following the strategy described previously[25]. A sequence logo was generated using WebLogo web

interface (http://weblogo.threeplusone.com)[72], which provides a graphical representation of the multiple sequence alignment, where each position's stack height indicates sequence conservation, and symbol heights within the stack reflect the relative frequency of individual amino acids at that position.

## Plasmid construction and rice transformation

DNA constructs were created with the Gateway cloning technology[73]. The genomic OsPSY1 and OsPSY2 sequences were amplified using the primers described in Supplementary Data 9. These sequences were then recombined with pENTR™/D-TOPO (Invitrogen, Cat#45-0218) to yield pTOPO_OsPSY1 and pTOPO_OsPSY2. The latter vectors were used in a Gateway LR cloning (Gateway® LR Clonase™ II Plus Enzyme Mix, Invitrogen; Cat#:12538-120) with pC1300-Ubi-Nos to yield *Ubi::OsPSY1* and *Ubi::OsPSY2* constructs. The generated vectors were transferred to *Agrobacterium tumefaciens* strain EHA105 and then introduced into the rice lines via Agrobacterium-mediated transformation. Rice transformation was performed following previously published protocols[74]. Post-transformant screening was based on hygromycin resistance and PCR determination of T-DNA insertion. Out of 52 positive lines for *Ubi::OsPSY1* and 53 for *Ubi::OsPSY2*, four homozygous lines were selected for characterization. Eventually, three homozygous lines, *Ubi::OsPSY1#9* (PSY1-9), *Ubi::OsPSY1#17*(PSY1-17), and *Ubi::OsPSY2#34* (PSY2-34), were selected for detailed studies.

## Gene expression analysis by quantitative PCR (qPCR)

Total RNA (1 mg) was extracted from the roots of seven-day-old rice seedlings grown hydroponically using TRIzol reagent (Invitrogen) and treated with the TURBO DNA-free kit (Ambion) to remove residual genomic DNA. cDNA was synthesized using the High-Capacity cDNA Reverse Transcription Kit (Applied Biosystems). The cycle threshold (Ct) value was measured on a Bio-Rad CFX96 Real-Time System coupled to a C1000 Thermal Cycler (Bio-Rad) using the iTaq Universal SYBR Green Supermix (Bio-Rad). Normalized relative quantities (NRQs) were obtained using the qBase method[75], with LOC_Os08g18110 (putative alpha-soluble NSF attachment protein) and LOC_Os11g26910 (putative SKP1-like protein 1B) as reference genes for normalization across samples[76]. NRQ values were normalized to the mean value obtained in wild-type plants. NRQ Melting curve analyses at the end of the process and "no template controls" were performed to ensure product-specific amplification without primer−dimer artifacts. Primer sequences are given in Supplementary Data 9. Three biological replicates were analyzed.

## Cellular analysis using LSCM

Laser scanning confocal microscopy (LSCM) was performed throughout the study using a Plan Apochromat ×63/1.40 oil and ×20/0.75 CS2 lens on a Leica TCS SP8 microscope.

Cell wall staining, seedling fixation and staining were performed using an adapted Clearsee protocol[77]. Briefly, the primary root of 7-day-old seedlings was sectioned into three short segments with a length of 0.5 cm starting from the root tip. The root segments were fixed for 1 h at room temperature in 10% neutral buffered formalin in PBS, using six-well plates, then washed five times for 1 min with PBS 1×. For root transverse section analysis, the 1 cm root fragments collected 4 cm from the root tip were embedded in 4% agarose and sectioned using a Leica automated vibrating blade microtome (VT1200 S). Once fixed, root fragments and transverse sections from those fragments were cleared in Clearsee solution for at least 24 hours under mild shaking. Fixed and cleared samples were incubated overnight in a Clearsee solution supplemented with 0.1% Calcofluor White. After 12 hours, the staining solution was removed, and samples were rinsed once in fresh Clearsee solution, then washed twice for at least 120 minutes in a renewed Clearsee solution with gentle shaking. Roots were carefully placed on a microscope slide with ClearSee and covered with a coverslip. Excitation and detection windows were set as follows: Calcofluor white for excitation at 405 nm, and detection between 415 and 570 nm. Central longitudinal section images were acquired from root complete fragments. Mature cortical cell length was assessed in 10 consecutive cells in the second cortical layer. Transverse section images were acquired to quantify the percentage aerenchyma formation, expressed as a percentage of the total cortical cell area.

## Soil collection and growth conditions in the greenhouse experiments

All data presented here were generated from two greenhouse studies carried out at the University of California−Davis in the fall of 2022 and the spring-summer of 2024. Soil from a rice field in Davis (38.543224 degrees North and -121.810536 West) was collected in July 2022 and in July and December 2023 using a front-end loader to gather down to a depth of ~6–10 inches. The soil at the field site belongs to the Capay soil series, a fine, smectitic, thermic Typic Haploxerert. Soil samples were analyzed at the UC Davis Analytical Lab for chemical content (Supplementary Data 10). Soil pH was 6.59 on average. For both experiments, all soils were transported back to the greenhouse, crushed to a 2 mm particle size, thoroughly homogenized, and stored until planting. Then, the soil was transferred into new 5.5 × 5.5 inch pots (2000 g per pot), which were then placed into five 23 gallon tubs per genotype (18 pots each). Prior to seedling transplantation, enough water was added to the tubs to submerge the soils.

Seeds were germinated in solid MS media, and root growth was followed as previously described for 7 days. Axenic seedlings were then transplanted into the watered pots (one seedling per pot) in the greenhouse, where they were irrigated with deionized water every other day to keep the soil submerged. Fertilized water [N, 77 ppm (parts per million); P, 20 ppm; K, 75 ppm; Ca, 27 ppm; Mg, 17 ppm; S, 68 ppm; Fe, 1.5 ppm; Cu, 0.02 ppm; Mn, 0.5 ppm; Mo, 0.01 ppm; Zn, 0.1 ppm; B, 0.5 ppm] replaced the distilled water in the tubs after microbiome samples were collected on day 15, 30, 55 and 70 after transferring to the pots. Each tub contained only one plant genotype to avoid root exudate compositional changes. We also kept a separate tub with unplanted pots and bulk soil controls. Tubs were distributed in the greenhouse benches following a completely randomized block design. All weeds were manually removed from the pots when identified.

Plant developmental stage and height were measured on the microbiome samples collection days. Roots and shoots were collected 100 days after transferring plants to pots, and dry biomass was measured. Seed number and one thousand seed weight were also calculated for each plant.

## Cell wall residue preparation

Lyophilized rice roots used for cell wall residue preparation method were adapted from Fonseca Garcia et al.[78]. Samples for lignin analysis were collected from 7-day-old rice seedlings or from root systems of 90-day-old plants grown in agricultural soil. Samples from older root systems were collected in the same root-shoot junction location as the one used for rhizosphere sample collection.

The fresh root biomass was lyophilized for 3 days, and then ball-milled biomass was extracted sequentially by sonication (15 min) with 80% ethanol (three times), acetone (one time), chloroform−methanol (1:1, v/v, one time) and acetone (one time).

## TGA lignin

Aliquots of 15 mg of cell wall residue were weighed into Eppendorf tubes and mixed with 1 ml of 3 N HCl and 0.1 ml thioglycolic acid (adapted after Suzuki et al.[79]). Samples were incubated at 80 °C for 4 hours and repeatedly mixed. Samples were rapidly cooled on ice and centrifuged for 10 min at 15,000 × g. The supernatant was discarded. Pellets were washed three times with distilled water (1 ml). Thereafter,

pellets were incubated with 1 ml of 1 N NaOH for 16 hours on a shaker at room temperature. The suspension was centrifuged for 10 min at 15,000 × $g$. The supernatant was carefully transferred into a 2-ml Eppendorf tube. The resulting supernatant was combined with the first alkaline supernatant and mixed with 0.2 ml concentrated HCl. Samples were incubated for 4 hours at 4 °C to precipitate the lignothioglycolate derivates. The samples were centrifuged, the supernatant discarded, and the pellet solubilized in 1 ml of 1 N NaOH. Absorbance of the resulting solution was measured at 280 nm. Results were expressed as an absorbance per mg cell wall.

## Measurement of methane emissions

Gas flux measurements of $CH_4$ were made in real-time in a transparent dynamic flow chamber using Picarro G2508 Gas Analyzer. In the fall 2022 experiment, fluxes at a total of four timepoints (15 d, 30 d, 55 d, 70 d) were measured across the growing season for 5 biological replicates corresponding to Kitaake, PSY1-17, PSY1-9, and bulk soil sample types. In the spring-summer 2024 experiment, $CH_4$ fluxes at a total of 12-timepoints were measured across the growing season for five biological replicates corresponding to Kitaake, PSY1-17, PSY2-34, and bulk soil sample types. Each replicate constituted a single rice plant except for bulk soil samples, which represented a negative control. Outlier detection was performed using the median absolute deviation (MAD) approach. Measurements deviating by more than three MADs from the median within a given time point and genotype were classified as outliers and excluded. This resulted in a reduced number of biological replicates for some timepoints. Because plants were destructively sampled for microbiome analyses at specific time-points, repeated measurements over time corresponded to the same tub rather than the same individual plant. Accordingly, area under the curve (AUC) was calculated for each tub by integrating methane flux values across time. At the start of every measurement, a gas-tight chamber was placed over a rice plant with minimal disturbance. The chamber was left on for at least 5 minutes for flux analysis. Gas fluxes were calculated by fitting linear regression lines across the measurement time window.

The conversion to mass fluxes using the Ideal Gas Law equation:

$$F = \frac{P \times V}{R \times T} \times \frac{\frac{\Delta CH_4}{\Delta t}}{A} \times 10^{-6}$$

where $F$ is the $CH_4$ flux (mg m$^{-2}$ h$^{-1}$), $P$ is the pressure, $V$ is the volume, and $A$ is the area of the sampling chamber. $R$ is the ideal gas constant, and $\Delta CH_4/\Delta t$ is the slope of the gas concentration change over time fitted by a linear equation. $T$ is the temperature in the sampling chamber ($K$).

## Microbiome sample collection

Rhizosphere samples were collected 15, 30, 55, and 70 days after rice was transplanted into pots. Briefly, we removed rice roots from pots and manually shook them to remove excess soil from the roots, leaving only ~1 mm of soil attached. These roots were transferred to 50-mL Falcon tubes that contained 15 mL of Powerbead Solution (Qiagen), stirred vigorously with sterile forceps to obtain closely attached soil, and frozen immediately in liquid nitrogen after removing roots. Samples were stored in a −80 °C freezer before nucleic acid extraction.

## Extraction and sequencing of total DNA and RNA

Approximately 2.0 g of rhizosphere soil was used for co-extraction of DNA and RNA by combining the Qiagen RNeasy PowerSoil Total RNA Kit and RNeasy PowerSoil DNA Elution Kit. Isolated nucleic acids were qualified and quantified using Nanodrop (Thermo Fisher) and Qubit (Invitrogen). For DNA samples, libraries were prepared using a Roche KAPA HyperPrep Kit and sequenced on Illumina NovaSeq6000 PE150 platform in Maryland Genomics. For RNA samples, ribosomal RNA

(rRNA) was first depleted using the NEBNext rRNA Depletion Kit, then libraries were prepared using NEBNext Ultra II Directional RNA Library Prep Kit and sequenced as above. In total, 72 DNA samples and 72 RNA samples (three replicates for each of the bulk soils and five replicates for each planted with the PSY1 rice genotypes and Kitaake control, at each time point) were sequenced, generating 974.8 Gbp and 920.7 Gbp of metagenomic and metatranscriptomic raw read data. Details for each sample can be found in Supplementary Data 1.

## Metaomics de novo assembly and binning

Raw 150-bp paired reads from metagenomic sequencing were processed using BBDuk contained in the BBTools suite (https://jgi.doe.gov/data-and-tools/software-tools/bbtools/). Biological replicates from the same samples were merged using BBMerge. Samples were individually assembled using metaSPAdes (v3.15.5) with default parameters[80]. Scaffolds longer than 1 kb were binned by a combination of MetaBAT2 (v2.15)[81], CONCOCT (v1.1.0)[82], MaxBin2 (v2.2.7)[83], and VAMB (v3.0.2)[84], using differential abundance calculated by pairwise cross-mapping against all samples by BBMap. Resulting bins from the same samples were integrated and optimized using DAS Tool (v1.1.2)[85]. Optimized bins derived from all samples were dereplicated at 98% whole-genome average nucleotide identity (ANI) using dRep (v3.4.0)[86], generating a set of approximately subspecies-level genomes. Quality of representative genomes was assessed using CheckM2 (v1.0.1)[87], and taxonomy was assigned by GTDB-Tk (v2.3.0)[88] based on the reference database version r214[89] and further converted to NCBI taxon. Only genomes with predicted completeness ≥50% and contamination ≤5% were used for downstream analyses ($n$ = 296).

For metatranscriptomics, raw 150-bp paired reads were pre-treated using BBDuk, and rRNA was filtered by SortMeRNA (v4.3.6)[90]. Clean reads primarily containing mRNA were individually assembled using rnaSPAdes (v3.15.5)[91] to generate transcripts.

## Gene prediction and annotation

Open reading frames (ORFs) in assembled scaffolds and transcripts were predicted by Prodigal (v2.6.3)[92]. Protein sequences from all samples were combined together and dereplicated at 95% identity along with 85% coverage to generate a non-redundant gene set, using Linclust integrated in MMseqs2[93,94]. Predicted genes were basically annotated against KEGG[95], UniRef100, and UniProt[96] databases by USEARCH v10[97], and against the CAZy database[98] using the online server following the manual instruction[99]. Protein domains were searched against Pfam (release 36.0)[100] using Astra (https://github.com/jwestrob/astra).

For the identification and clustering of *rpS3* genes, the PF00189 HMM was first used to search against predicted proteins based on pre-built noise cutoffs using HMMER (v3.3)[101]. Corresponding nucleotide sequences of protein hits were pulled out and clustered at 99% ident[102] using VSEARCH (v2.13.3)[103]. Representative sequences from each cluster were blasted against the Genbank nr database to get preliminary taxonomic information[104]. Precise taxonomy was further retrieved by phylogeny construction that involved alignment of sequences using MAFFT (v7.453)[105], trimming of alignments using tri-mAl (v1.4.rev15)[106], and generation of phylogenetic trees using IQ-TREE v1.6.12 with best-fit models automatically determined[107]. Trees were decorated on the iTOL server for better visualization[108].

For genes closely related to $CH_4$ cycling, the PF02249 and PF02745 HMMs were used to recruit McrA-like sequences; the TIGR03079 was used to recruit PmoB-like sequences; and the PF02964 was used to recruit the gamma subunit of methane monooxygenase hydroxylase. Further identification and classification were performed based on phylogeny as described above.

For genes involved in $H_2$ production and consumption, three curated HMMs were used to target sequences homologous to [NiFe], [FeFe], and [Fe] hydrogenases, respectively[42]. Groups and subtypes of

identified hydrogenases were distinguished based on protein phylogeny. [FeFe] Group A1 hydrogenases were further classified using the online tool HydDB, which integrates genetic context for accurate classification[42]. For genes related to nitrate reduction, proteins that contain the molybdopterin-binding domain or molybdopterin oxidoreductase domain were phylogenetically classified into distinct groups, including NarG and NapA exclusive groups. NifH proteins were identified using the TIGR01287 HMM that targets nitrogenase iron protein, included in all three nitrogenase forms (i.e., Mo-Fe, V-Fe, and Fe-only). Reductive DsrA was identified by DiSCo (v1.0.0)[109]. Taxonomy of genes was identical to that of gene-located scaffolds, which was assigned if 50% of the co-existing proteins belonged to the same taxon.

### Abundances of assembled genomes and genes

Metagenomic abundances of genomes and genes were calculated using CoverM (v0.6.1)[110]. Given genomes dereplicated at 98% ANI, a minimum read identity during mapping was set at 98% together with a minimum aligned per cent of 75% in the "genome" mode. For non-redundant genes at a 95% identity level, corresponding scaffolds were mapped by reads with a minimum identity of 95% and an alignment percentage of 75% using the "contig" mode. Abundances were considered to be zero if the covered fraction by any read was <50%. Normalized abundances were calculated by dividing coverage by total read length and multiplying by a factor of 1E11 for visualization.

Overall metatranscriptomic abundances of genomes were calculated using CoverM with the same cutoff parameters as above, followed by a normalization in which aligned read counts for each genome were divided by total read size and genome length, and multiplied by a factor of 1E15. To calculate the abundances of genes, corresponding scaffolds were first mapped by metatranscriptomic reads using Bowtie2 with "--end-to-end" "--sensitive" mode[111]. Exactly aligned reads for genes inside scaffolds were counted using featureCounts (v2.0.6)[112]. Final abundances, that said transcriptional activities, were normalized by dividing read counts by gene length and multiplying a factor of 1E11 for visualization.

### Enrichment of genes/pathways in upregulated genomes

Based on calculated overall transcriptional abundances of genomes, DESeq2[113] was used to identify significant differences in genome activities between the Kitaake and PSY1 genotypes, and between rice planted and unplanted groups. We consider genomes as "upregulated genomes" if: 1) they had significantly higher overall activities (log2-FoldChange ≥ 1, $p \leq 0.05$) in PSY1 rice genotypes compared with the Kitaake in ≥1 rice growth stage; 2) they never showed significantly higher overall activities in the Kitaake than PSY1 genotypes; and 3) they were more active in rice planted groups than in bulk soils. Finally, 17 of 296 genomes were identified as up-regulated genomes in PSY1 genotypes.

KEGG Orthology (KO) identifiers were assigned to predicted proteins in all genomes, using kofamscan (v1.3.0; https://github.com/takaram/kofam_scan) with KOfam set r2023-10-02. Only hits above score thresholds and with the highest score were selected for individual proteins. Next, we performed Fisher's exact test to evaluate the over/under enrichment of identified KOs in up-regulated genomes ($n = 17$) relative to remaining genomes ($n = 279$). A total of 3509 KOs that appeared in ≥10 genomes were tested, and the resulting $p$ values were adjusted using Benjamini–Hochberg method[114]. KOs with FDR ≤ 0.05 were considered significantly enriched (odds ratio > 1) or depleted (odds ratio <1) in up-regulated genomes.

Given KEGG metabolic pathways comprised of multiple KOs, we quantified pathways by summing up the abundances of all genes that are associated with related KOs. Differential abundances were calculated by subtracting abundances in the PSY1 genotypes from abundances in the Kitaake, shown as PSY1-17 and PSY1-9, respectively.

### Calculations of genome-wide SAP values

We followed the previous paper to predict SAP values from genomes[46]. Briefly, we first calculated the relative abundances (number of genes normalized by the total number of genes in genomes) of sugar and acid genes that are implicated in the corresponding degradation pathways, based on the list in the above reference. The generalized linear model was then used to predict SAP values for each genome, as:

$$SAP = tanh(60.76 \times S - 20.21 \times A)$$

where S and A indicate relative abundances of sugar and acid genes, respectively. Statistical difference in average SAP values between genome groups was calculated using the Wilcoxon rank sum test.

### Rice roots and exudate collection

Samples for metabolomic analysis were collected from 7-day-old rice seedlings grown vertically on 1× MS solid media plates. Prior to collection, seedlings were removed from the plates and thoroughly washed with deionized water to remove any residual media from the roots. For root tissue samples, the entire root system of 20 seedlings was ground using a mortar and pestle, with a subsample transferred to a 2 mL tube (Eppendorf #022363352). Tissue fresh weight was recorded with pre-weighted tubes, and samples were stored at −80 °C until further processing. For root exudate samples, 15 seedlings were placed in 50 mL flasks containing ultra-pure water and held in place by sponges so that only the root tips were submerged. The flasks were incubated with shaking for 2 hours before the exudates were collected in 50 mL tubes (Fisher Scientific#14-959-49 A). Exudates were initially stored at −20 °C for a few hours, then transferred to −80 °C until further use.

### Liquid chromatography sample preparation

To extract metabolites from 300 to 400 mg root tissue and agar control media, samples were first frozen, lyophilized dry (FreeZone 2.5 Plus; Labconco), then powderized by bead-beating (Mini-Beadbeater96, BioSpec) for 5 seconds, 2×, using a 3.2 mm stainless steel bead. Next, 500 µL of 100% methanol was added to each powderized root sample and 1.5 mL to each powderized agar sample, briefly vortexed, then sonicated in an ice water bath for 10 minutes. Samples were centrifuged (5 min, 11,000 × g) to pellet debris, supernatant removed, and transferred to a 2 mL Eppendorf, dried in a SpeedVac (SPD111V; Thermo Fisher Scientific). To extract metabolites from exudates collected in 40 mL water, samples were first frozen then lyophilized dry. Next, 3 mL of 100% methanol was added to each sample, briefly vortexed, then sonicated in an iced water bath for 10 minutes. Samples were centrifuged (5 min, 7000 × g) to pellet debris, supernatant removed and transferred to a 5 mL Eppendorf, then dried in a SpeedVac. Next, 1 mL of 100% methanol was added to each dried extract, briefly vortexed, then sonicated in an ice water bath for 10 minutes. Samples were centrifuged again, then supernatant removed and transferred to a 2 mL Eppendorf and dried in a SpeedVac. All dried extracts were stored frozen at −80 °C. The same extraction procedures were followed using empty 50 mL and 2 mL tubes to serve as extraction controls, and a 50 mL tube containing 50 mL water to serve as an exudate collection control (3 replicates each).

In preparation for LC-MS, samples were resuspended in 100% methanol containing a mix of isotopically labeled internal standards (https://doi.org/10.17504/protocols.io.kxygxydwkl8j/v1). Resuspension volume was varied for each root extract to normalize by amount of extracted biomass (1500 mg/mL), while all other samples were resuspended in 150 µL.

### LC-MS/MS metabolomics analysis

Samples were run using both normal and reverse-phase chromatography. Each of these was performed using an Agilent 1290 LC stack, with MS and MS/MS data collected using an Exploris 120 Orbitrap MS

(Thermo Fisher Scientific, San Jose, CA) for normal phase, and QExactive HF Orbitrap for reverse phase. Mass spectrometer parameters and gradients are provided in this protocols.io (https://doi.org/10.17504/protocols.io.kxygxydwkl8j/v1) for normal phase, and this protocols.io (https://doi.org/10.17504/protocols.io.ewov19r27lr2/v1) for reverse phase. For both normal phase and reverse phase, full MS1 spectra were collected at 60k resolution in both positive and negative ionization mode, with MS/MS fragmentation data acquired using stepped then averaged 10, 20, and 40 eV collision energies at 15 k resolution. For normal phase, the MS1 m/z range for data collection was 70-1050, while for reverse phase, both m/z 80–1200 and m/z 1000–3000 were used in separate runs to cover mass ranges for metabolites and peptides. Samples consisted of three biological replicates each and extraction controls, with sample injection order randomized by replicate and an injection blank of 100% methanol run between each sample, with the blank replaced by an injection of internal standard mix every third sample, as well as QC mix every 15 samples. The raw data from the metabolite LC-MS runs are provided. For untargeted data analysis, features with 0.8 min<RT < 18.5 min (HILIC column) or features with 0.5 min<RT < 9.5 min (C18 column), and maximum peak height fold-change between sample and extraction control >3 were selected. We performed a Feature-Based Molecular Networking (FBMN)[115] workflow using MZmine-3.7.2[116] and a Global Natural Products Social Molecular Networking (GNPS2; http://gnps2.org)[117]. The MZmine workflow was used to generate a list of features obtained from extracted ion chromatograms containing chromatographic peaks within a narrow m/z range and filtered to remove isotopes. For each feature, the most intense fragmentation spectrum was uploaded to GNPS for putative identification by comparison with mass spectra deposited in the database. Compound classes were attributed to identified compounds using NPClassifier[118].

For targeted analysis, compound identification was performed by comparing detected *m/z*, retention time (RT) and fragmentation pattern to standards run in-house using the same LC-MS methods and assigning a confidence level based on these criteria. For each identified compound, the highest level of confidence (Exceeds Level 1)[119] was achieved when the measured m/z agreed with the expected theoretical m/z with less than 5 ppm error, detected RT of the feature was within 0.5 minutes of theoretical, and the fragmentation pattern matched that of the standard. When MSMS was not collected, some compounds were identified based on only mz and RT (Level 1); mis-matching MSMS would invalidate an identification.

## NPOC measurement
Frozen root exudates collected as described previously were thawed at 4 °C and then diluted to 8 mL using Milli-Q water. Samples were analyzed on a Shimadzu TOC-L CPH total organic carbon analyzer equipped with TNM-L and ASI-L modules using UHP synthetic air mix as carrier gas. Calibration curves for non-purgeable organic carbon were prepared using potassium hydrogen phthalate. Standards and samples were analyzed under the same settings; blanks and control samples were included for quality control purposes. Sample measurements were set up with 1 wash step, followed by automated acid addition at 1.5% 1 M HCl with a 2.5-minute sparging period at 80 mL/min to remove inorganic carbon and purgeable carbon. Acidified, sparged samples containing non-purgeable organic carbon were then injected at 50 μL for analysis using a max 4.5 min integration window for each carbon measurement; this was performed with 3–5 technical replicate injections with the max cv set at 2%. Measurements were corrected for dilution factors and then reported as ppm C (mg of C per L of exudate).

## Soil incubations with rice root exudates
To assess the direct impact of rice root exudates on $CH_4$ emissions, microcosm incubations were performed in wide-mouth glass mason jars (volume: 500 mL) fitted with gas-tight rubber septa. Each microcosm was prepared by mixing 10 g of air-dried agronomic soil, collected from the same rice basin used in the greenhouse experiments, with 10 mL of deionized water. The microcosms were incubated at 28 °C in the dark for 48 hours to allow microbial respiration to stabilize. Root exudates from Kitaake, PSY1-, and PSY2-overexpressing rice seedlings were collected as previously described, lyophilized, and reconstituted in 1 mL of Milli-Q water. Each microcosm was flushed with nitrogen gas for 1 minute to displace ambient oxygen, and then exudates were gently added to the soil, taking care to avoid physical disturbance. The jars were sealed immediately, and headspace gas samples were collected at 0, 5, 10, 20, 30, 60, 120, 180, 240, and 300 minutes post-addition. $CH_4$ concentrations in the headspace were quantified using a cavity ring-down spectrometer (Picarro G2201, Picarro Inc.), providing a direct measurement of microbial $CH_4$ emissions in response to rice root exudate chemistry. $CH_4$ concentrations in ppm were converted to nanograms of $CH_4$ using the ideal gas law, assuming a constant headspace volume of 500 mL throughout the experiment, with final values expressed in mg $CH_4$ per g of dried soil.

## Construction and analysis of genome-scale metabolic models
We selected microbial genomes for metabolic modeling based on the following criteria: (1) genomes contain $H_2$-producing or $H_2$-consuming hydrogenases; (2) genomes are predicted to have completeness > 90% and contamination <5%; (3) genomes are affiliated with different phyla. Three $H_2$-producing genomes from Bacteroidia, Ignavibacteriota, and Moranbacteria, and three $H_2$-consuming genomes from Acidobacteriota, Proteobacteria, and Deinococcota were finally selected and annotated using the App "Annotate Genome/Assembly with RASTtk" (v1.073) in KBase[120]. The resulting annotated genomes were used as inputs for the App "MS2 - Build Prokaryotic Metabolic Models with OMEGGA" (v.2.0.0) to build gap-filled metabolic models using media comprised of basic nutrients and PSY1 over-secreted compounds (Supplementary Data 8). Basic nutrients included glucose and glutamate as primary carbon sources, which were present and had no significant difference in concentrations across rice genotypes. Also included were standard essential factors for microbial growth (e.g., $Fe^{2+}$). PSY1 over-secreted compounds were measured at significantly higher concentrations in root exudates from PSY1 rice genotypes compared with the Kitaake.

Dynamic flux balance analysis (dFBA) was performed for each microbial model individually, assuming well-mixed batch growth conditions, using the Computation Of Microbial Ecosystems in Time and Space (COMETS) platform (v2.12.2)[121]. We set the initial biomass at $1 \times 10^{-6}$ g and used default simulation parameters. To compare results across varying conditions, we calculated the maximal potential of substrate turnover and biomass yield when microbial growth stopped and substrates no longer changed, for each condition. As glucose and glutamate in basic nutrients were likely the most abundant carbohydrate and amino acid secreted by rice roots[122,123], concentrations of PSY1 over-secreted compounds were varied from 0.01–1-fold as that of basic nutrients. $O_2$ is transported to the rhizosphere through rice aerenchyma, and its concentrations were set at 1–10 fold that of basic nutrients. $H_2$ concentrations were set at 0 when testing $H_2$ production potential and at 10-fold when testing $H_2$ consumption potential. For visualization purposes, we calculated the consumed $H_2$ amounts by subtracting the final values from the initial.

## Statistical analysis
NMDS analyses for the rpS3 gene were performed and plotted using "vegan" and "ggplot2" packages in R (v4.2.2)[124]. Differences and significance between groups were assessed by analysis of similarities and permutational multivariate analysis of variance tests. Outliers were detected and removed from replicates using the MAD test[125]. Differential expressions of genomes between samples were calculated by DESeq2[113].

## Reporting summary

Further information on research design is available in the Nature Portfolio Reporting Summary linked to this article.

## Data availability

All metagenomic and metatranscriptomic sequencing reads have been deposited in the NCBI Sequence Read Archive (SRA) with the project accession number: PRJNA1161606. Microbial metabolic models have been deposited in figshare (https://doi.org/10.6084/m9.figshare.29115692). Assembled microbial genomes are present in NCBI Bio-Project PRJNA1161606. They can also be accessed at: https://ggkbase.berkeley.edu/Assembled_genomes_from_rice_genotype_experiment. The metabolomics raw data can be accessed at: https://massive.ucsd.edu/ProteoSAFe/dataset.jsp?task=3349af8926624635af24c6b01babebd1. Source data are provided with this paper.

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

## Acknowledgements

We would like to thank Ryan C. Packer, Christina Fossum, and Rory M. Greenhalgh for their help setting up the greenhouse experiment. We thank Bethany C. Kolody and Jackie Zorz for assistance in using R, and

Daniel A. Gittins for help with hydrogenase HMMs. We thank Leylen Miloslavich for help with soil incubation experiments. This project has been made possible in part by grant number CZIF2022-007203 from the Chan Zuckerberg Initiative Foundation (to J.F.B., P.C.R., and J.P.R.), and by support from the Emerson Collective (J.F.B.). This material is based upon work supported by the Joint BioEnergy Institute, U.S. Department of Energy, Office of Science, Biological and Environmental Research Program under Award Number DE-AC02-05CH11231 with Lawrence Berkeley National Laboratory (P.C.R. and H.V.S.). Funding was also provided by the Bill and Melinda Gates Foundation (grant number INV-037174 to J.F.B.). The findings and conclusions are those of the authors and do not necessarily reflect positions or policies of the Bill and Melinda Gates Foundation. The metabolomic analysis was conducted by the U.S. Department of Energy Joint Genome Institute (https://ror.org/04xm1d337), a DOE Office of Science User Facility supported by the Office of Science of the U.S. Department of Energy, operated under Contract No. DE-AC02-05CH11231 (T.R.N.). A.M.S. is supported by a U.S. Department of Agriculture (USDA) National Institute for Food and Agriculture (NIFA) Postdoctoral Fellowship (2023-67012-39889). Work conducted by J.P.R. and K.E.M. was supported under the auspices of the DOE, LLNL Contract DE-AC52-07NA27344, and DOE BER award SCW1632. This material by m-CAFEs Microbial Community Analysis & Functional Evaluation in Soils (https://mcafes.lbl.gov), a Science Focus Area led by Lawrence Berkeley National Laboratory, is based upon work supported by the U.S. Department of Energy, Office of Science, Office of Biological & Environmental Research under contract number DE-AC02-05CH11231(S.M.K., D.S. and T.R.N.).

## Author contributions

The study was conceived by L.D.S., M.F.E., P.C.R., and J.F.B. M.F.E., S.S., A.T.A.J., T.S.W., and A.M.S. processed the soil from rice paddies at UC Davis, planted rice in the greenhouse, and performed phenotypic analyses. L.D.S. and J.K. measured $CH_4$ flux and collected rhizosphere soil samples with the help of M.F.E., A.T.A.J., A.M.S., S.S., and T.S.W. L.D.S. extracted DNA and RNA from collected soil samples and sequenced them and performed all metagenomic and metatranscriptomic analyses. M.F.E., T.S.W. and A.T.A.J. collected rice root tissue and exudate samples for metabolomic analysis. T.R.N., K.B.L. and B.P.B. designed the metabolomics study, generated and processed the metabolomic data, and M.F.E. analyzed the metabolomic data. S.M.K. prepared the NPOC measurements. H.T. performed the lignin analysis in young and adult rice plants. H.V.S. supervised cell wall analysis. L.D.S. constructed microbial metabolic models and performed dynamic flux balance analysis with the help of I.D. and D.S. R.S. provided technical support in processing sequencing data. L.D.S., M.F.E., and K.E.M. performed soil incubation experiments with the help of A.T.A.J. and supervised by J.P.R. L.D.S., M.F.E., P.C.R. and J.F.B. wrote the manuscript with input from all the authors.

## Competing interests

J.F.B. is a co-founder of Metagenomi. T.R.N. is an advisor to Brightseed Bio. The remaining authors declare no competing interests.

## Additional information

[1]Innovative Genomics Institute, University of California, Berkeley, CA, USA. [2]Department of Plant Pathology and the Genome Center, University of California, Davis, CA, USA. [3]Environmental Science, Policy and Management, University of California, Berkeley, CA, USA. [4]Physical and Life Sciences Directorate, Lawrence Livermore National Laboratory, Livermore, CA, USA. [5]Bioinformatics Program, Faculty of Computing and Data Sciences, Boston University, Boston, MA, USA. [6]Biological Design Center, Boston University, Boston, MA, USA. [7]Center for Advanced Interdisciplinary Research, Ss. Cyril and Methodius University, Skopje, North Macedonia. [8]Feedstocks Division, Joint BioEnergy Institute, Emeryville, CA, USA. [9]Environmental Genomics and Systems Biology Division, Lawrence Berkeley National Laboratory, Berkeley, CA, USA. [10]Department of Energy System Engineering, Karadeniz Technical University, Trabzon, Turkey. [11]Joint Genome Institute, Lawrence Berkeley National Laboratory, Berkeley, CA, USA. [12]Department of Plant and Microbial Biology, University of California, Berkeley, Berkeley, CA, USA. [13]Life and Environmental Sciences Department, University of California Merced, Merced, CA, USA. [14]Department of Physics, Boston University, Boston, MA, USA. [15]Department of Biomedical Engineering, Boston University, Boston, MA, USA. [16]Department of Biology, Boston University, Boston, MA, USA. [17]Earth and Planetary Science, University of California, Berkeley, CA, USA. [18]These authors contributed equally: Ling-Dong Shi, Maria Florencia Ercoli. ✉e-mail: pcronald@ucdavis.edu; jbanfield@berkeley.edu

