## [Transparent Peer Review file · Nature Communications]

Reduced methane emissions in transgenic rice genotypes are associated with altered rhizosphere microbial hydrogen cycling

Corresponding Author: Professor Jillian Banfield

Version 0:

Reviewer comments:

Reviewer #1

(Remarks to the Author)

Review of Manuscript NCOMMS-24-78529

Title: Reduced methane emissions in transgenic rice genotypes are associated with altered rhizosphere microbial hydrogen cycling.

By: Shi et al.

The manuscript demonstrates the potential functions of PSY in controlling CH₄ emissions that arise from rice cultivation. The authors present data that links PSY to CH₄ mitigation through metagenomics, metatranscriptomic and metabolomics analysis. Considering that rice cultivation is a sector of crop production that contributes approximately 10% to the global CH₄ budget, mitigation practices are one strategy that may have great potential to mitigate the climate crisis.

The manuscript provides valuable insights into the factors that influence methane emissions from rice cultivations. Still, some parts of the manuscript need further clarification before it is ready for publication. Please see my comments below.

General comments:

1. Most of the conclusions concerning microbial activities come from different omics techniques. Preferentially, those should be complemented with functional experiments that demonstrate the ability of the suggested bacteria to, e.g. produce or consume hydrogen.
2. The authors present data from Greenhouse experiments. The data presented demonstrate that PSY can influence CH₄ emissions. However, it is not certain that PSY will do so under field conditions since different soil and temperature conditions influence methane emissions from rice cultivations. Preferentially, the authors should complement the methane emission measurements under Greenhouse conditions with field trial measurements to prove that the reduction in methane emissions is a general phenomenon.

Specific comments:

3. Line 104-106. I don't think the phylogenetic tree displayed in Supplemental Figure 1 b illustrates the claim that the authors make, and I lack a description of how the phylogeny was done. Is only the conserved domain included in the analysis? As far as I can tell, this analysis is not mentioned in the materials and methods section. If the authors want to compare the conserved domain, maybe this could be better shown with a sequence alignment.
4. Line 116-117. The authors hypothesize that LRR-RLKs may interact with the applied synthetic OsPSY peptides in root tissues and that this could influence root growth. Please provide an explanation for why.
5. Line 143-143. The authors detect lower lignin content in PSY1-17 lines compared with Kitaake but not in the other transgenic lines. Please provide an explanation for this discrepancy between the phenotypes of the transgenic lines.
6. Line 166-188. The structure of the text is difficult to follow since the results for EXP2 are presented before EXP1. I think

the authors should restructure the text.

7. Line 212-219. The style of writing in this section resembles a Materials and Methods section. Please reformulate.

8. Line 220. The authors base their taxonomy on assembled rps3 sequences. It has recently been shown that metagenome assemblies frequently fail to detect ribosomal sequences (Mise, K., Iwasaki, W. Unexpected absence of ribosomal protein genes from metagenome-assembled genomes (ISME COMMUN. 2, 118 (2022). <https://doi.org/10.1038/s43705-022-00204-6>). The authors should revise their taxonomic analysis accordingly and include more sequences as a base for their taxonomic classification.

9. Line 232-234. The text in the manuscript does not match the text in the legend of Figure 3d:

Text in the manuscript: ...cultivation time has a large and statistically significant effect on the community structure (dissimilarity score of up to 0.539) but rice genotype does not (Fig. 3d)

Text in the Figure legend: ...for community structure (d), $R=0.539$, $P=0.001$ between rice genotypes (highlighted by ovals), and $R=0.0729$, $P=0.238$ between cultivation time;

10. Line 287-290. This is a strong statement since the authors base their conclusion on transcriptome data, not functional microbiome studies. The authors should acknowledge this fact.

11. Line 301. Root secretion is an end product of root and soil microbial activity. The authors use axenic rice roots in their metabolic analyses, which may differ from real-life rhizospheric conditions. The authors should acknowledge this in the text.

12. Line 304-310. I think the authors could help the reader here by rearranging either the text in the manuscript or the figures. The authors write "root tissues or exudates" in the text and refer to Fig S5. a and e. However, Fig S5 a shows data for root exudates, and Fig S5 e shows data for root tissues. The remaining text in this paragraph follows a similar pattern. This makes the text unnecessarily difficult to follow.

13. Line 355-378. The metabolic models need further clarifications:

- a. The authors use PSY over secreted compounds in their models. How do the models look if the corresponding over-secreted compounds from the wild-type comparator were used?
- b. Also, does the total amount of secreted compounds differ between the genotypes or is only the composition. If there is an increased total amount of compounds released, this will likely have a large effect on methane emissions.
- c. The authors identify 17 microbial genomes with increased transcriptional activities in soils of PSY genotypes of which five had H₂-consuming hydrogenases, please provide information on the taxonomy of these microbial genomes.
- d. The authors then build the models on three of these microbial genomes. How were these three genomes selected? Which genomes of the 17 were used, and what were the species? All this information is needed to evaluate the models.
- e. Line 375-378. The final sentence should be clarified since it conveys conflicting results.
- f. Finally, if the species used in the models can be cultivated, it should also be possible to verify the models experimentally.

14. Line 398-404. The authors state that methane cannot be derived from acetate as low activity of known aceticlastic methanogens was found, and they suggest that acetate might not even be produced or only be present at low levels. Acetate, together with H₂, is the final products from degradation of all organic compounds in an anaerobic environment, and thus, it seems very unlikely that no acetate is produced. Moreover, the authors do not mention the possibility of the alternative route for methane production from acetate, that is syntrophic acetate oxidation (SAO). This route for methane production has been found to be present in rice soil and is likely to be one pathway for hydrogen production feeding hydrogenotrophic methanogens. The text has to be revised accordingly. Moreover, the activity of this group would also be relevant to explore in the transcriptomic data, e.g. the WL pathway, possibly combined with the glycine cleavage system recently shown important for SAO. This pathway is also relevant for H₂ oxidizing acetogens, possibly representing an important H₂ oxidizing bacterial group, in addition to sulfate, iron and reducing bacteria.

Reviewer #2

(Remarks to the Author)

Reviewer #3

(Remarks to the Author)

Overall comments

1. The manuscript is well-structured and provides a comprehensive analysis demonstrating the methane emission mitigation effect of PSY in rice. However, the metagenomic and metabolomic data interpretations should provide more concrete evidence of causality rather than correlation. For example, while PSY-expressed plants show altered exudates, how do we confirm that these exudates are indeed directly responsible for microbial shifts?

2. The microbial community analysis (Figure 3) could benefit from clearer labeling—perhaps grouping similar microbial taxa together visually.

3. The authors propose that PSY exudates selectively enhance H₂-oxidizing microbes while suppressing H₂-producing ones. This is intriguing, but it is unclear whether:

- This effect is directly caused by specific exudate compounds or just root-associated environmental changes.
- The microbial shifts persist beyond controlled conditions.

-The observed metabolic effects (e.g., gluconeogenic acid enrichment) are due to PSY expression or indirect plant growth effects.

4. What are the agronomic trade-offs of PSY-expressing rice? The paper mentions shorter stature, reduced grain weight, and other agronomic traits. How might these factors impact yield? Similarly, Iqbal et al. (2021) demonstrated that while methane emissions were significantly reduced, there was a trade-off with rice yield. Both OsLSD (Iqbal et al.) and OsPSY influence root biomass and aerenchyma development, which typically leads to reduced rice growth..

Iqbal, M. F., Liu, S., Zhu, J., Zhao, L., Qi, T., Liang, J., ... & Fan, X. (2021). Limited aerenchyma reduces oxygen diffusion and methane emission in paddy. *Journal of Environmental Management*, 279, 111583.

5. Is it differently expressed PSY on other parts? (panicle, leaf etc)

6. PSY decrease lignin contents, please provide correlation with lignin and methane emission.

7. How do the haplotypes of the PSY gene appear in natural variants, and what are their potential effects?

Detail comments

-p. 16 Line 1: It is not appropriate to compare the loss-of-function GS3 allele with PSY overexpression, as GS3 is a natural variant, whereas PSY is an overexpressed mutant. Therefore, the methane mitigation effect should be compared with other gene-edited varieties or genes.

-(p. 5 Line : 90 Introduction) Please provide sufficient information about PSY gene and role of PSY1 and PSY2

-(p. 23 Line : 654 Methods) Please provide how many bp read to analyze metagenomics analysis (rpS3)

Version 1:

Reviewer comments:

Reviewer #1

(Remarks to the Author)

Review of Manuscript NCOMMS-24.785229

I thank the authors for their updated and improved manuscript. The authors have provided tangible answers to most of my comments and concerns regarding the original submission. I only have a few additional comments:

1. I note that the authors have chosen not to perform field trials with their transgenic plants. While such experiments would be informative, acknowledging this limitation suffices, given the quality of the data now presented in the manuscript.

2. The authors do not understand my reference to "functional microbiome studies" (comment 10 in the first round of revision). Functional microbiome studies would include cultivation of microbiome isolates and studies of H₂ production and CH₄ emissions under controlled conditions. As noted by the authors in lines 524-524, such experiments could be used to further test and substantiate the models presented in Figure 7. In this context, it is also important to acknowledge the limitations of transcriptome studies and that they are not the same as functional studies. However, I think the text edits the authors have now made clarify the limitations of their study sufficiently.

Reviewer #2

(Remarks to the Author)

Reviewer #1 (Remarks to the Author):

Review of Manuscript NCOMMS-24-78529

Title: Reduced methane emissions in transgenic rice genotypes are associated with altered rhizosphere microbial hydrogen cycling.

By: Shi et al.

The manuscript demonstrates the potential functions of PSY in controlling CH₄ emissions that arise from rice cultivation. The authors present data that links PSY to CH₄ mitigation through metagenomics, metatranscriptomic and metabolomics analysis. Considering that rice cultivation is a sector of crop production that contributes approximately 10% to the global CH₄ budget, mitigation practices are one strategy that may have great potential to mitigate the climate crisis.

The manuscript provides valuable insights into the factors that influence methane emissions from rice cultivations. Still, some parts of the manuscript need further clarification before it is ready for publication. Please see my comments below.

Response: We thank the reviewer for their careful reading of the manuscript and appreciation of the significance of this work. To improve the manuscript, we have revised the text, figures, and tables as requested, and below provides point-by-point responses to each comment. Reviewers' comments are in normal text, followed by our response in dark blue. Dark blue font is also used to provide the revised manuscript text.

General comments:

1. Most of the conclusions concerning microbial activities come from different omics techniques. Preferentially, those should be complemented with functional experiments that demonstrate the ability of the suggested bacteria to, e.g. produce or consume hydrogen.

Response: We appreciate the reviewer's constructive suggestions. To further support our findings, we performed additional experiments where we incubated agronomic soil with root exudates collected from different rice genotypes and monitored CH₄ emissions over time.

While directly measuring hydrogen dynamics, as the reviewer suggested, would be an ideal way to validate our model, the rapid onset of CH₄ production and consumption upon exudate addition indicates that hydrogen turnover occurs on a timescale that is likely too fast for reliable detection under our experimental conditions. This is in agreement with previous observations that hydrogen is rapidly consumed by metabolic processes and

maintained at very low steady-state partial pressures (e.g., DOI: 10.1007/BF02341425). Despite this limitation, our results clearly demonstrate a direct and reproducible effect of exudates on microbial activity in the soil, independent of other plant phenotypes. These findings provide additional support for the model proposed in the manuscript and the conclusions drawn from our omics data.

Manuscript now reads as (lines 356-366):

To determine whether the altered exudate composition of PSY-genotypes can directly influence microbial activity, we conducted soil incubation assays using paddy soil from the same rice field used in our greenhouse experiments. Exudates collected from PSY1- and PSY2-overexpressing rice seedlings were added as amendments, with exudates from Kitaake and sterile water as controls. CH₄ emissions were then monitored over time. No CH₄ flux was detected in water-treated soils, confirming that exudates were necessary to stimulate methanogenesis. Soils treated with Kitaake exudates exhibited consistently higher CH₄ emissions than those treated with PSY1 or PSY2 exudates, mirroring trends observed in the greenhouse-grown plant experiments (Supplementary Fig. 11).

Supplementary Figure 11. Soil incubation with exudates from PSY-overexpressing and control rice genotypes. Agronomic paddy soil was incubated under anaerobic conditions with root exudates collected from PSY1- and PSY2-overexpressing seedlings or from wild-type Kitaake. A sterile water control was also included. Cumulative CH₄ emissions were measured over time. CH₄ emissions were normalized by grams of dry weight soil, as NPOC was comparable between different exudate samples.

Materials and methods now read as (lines 955-974):

Soil incubations with rice root exudates

To assess the direct impact of rice root exudates on CH₄ emissions, microcosm incubations were performed in wide-mouth glass mason jars (volume: 500 mL) fitted with gas-tight rubber septa. Each microcosm was prepared by mixing 10 g of air-dried agronomic soil — collected from the same rice basin used in the greenhouse experiments — with 10 mL of deionized water. The microcosms were incubated at 28 °C in the dark for 48 hours to allow microbial respiration to stabilize. Root exudates from Kitaake, PSY1-, and PSY2-overexpressing rice seedlings were collected as previously described, lyophilized, and reconstituted in 1 mL of Milli-Q water. Each microcosm was flushed with nitrogen gas for 1 minute to displace ambient oxygen, and then exudates were gently added to the soil, taking care to avoid physical disturbance. The jars were sealed immediately, and headspace gas samples were collected at 0, 5, 10, 20, 30, 60, 120, 180, 240, and 300 minutes post-addition. CH₄ concentrations in the headspace were quantified using a cavity ring-down spectrometer (Picarro G2201, Picarro Inc.), providing a direct measurement of microbial CH₄ emissions in response to rice root exudate chemistry. CH₄ concentrations in ppm were converted to nanograms of CH₄ using the ideal gas law, assuming a constant headspace volume of 500 mL throughout the experiment, with final values expressed in mg CH₄ per g of dried soil.

2. The authors present data from Greenhouse experiments. The data presented demonstrate that PSY can influence CH₄ emissions. However, it is not certain that PSY will do so under field conditions since different soil and temperature conditions influence methane emissions from rice cultivations. Preferentially, the authors should complement the methane emission measurements under Greenhouse conditions with field trial measurements to prove that the reduction in methane emissions is a general phenomenon.

Response: We thank the reviewer for highlighting the importance of validating our findings under field conditions. We fully agree that such confirmation is essential to establish the broader relevance of our greenhouse-based results. However, conducting a field study — particularly one that incorporates the same level of multi-omics characterization presented here — would require substantial time and adherence to regulatory requirements associated with the cultivation of transgenic plants. In our case, a field trial would take a minimum of 18 months to complete, due to the seasonal nature of planting, the need for multi-time point sampling, and post-harvest analyses. We have added a statement to the Discussion to acknowledge this limitation and to describe our plans for upcoming field-based validation studies.

Manuscript now reads as (lines 538-549):

While our greenhouse-based experiments using agronomic soil collected from a rice basin provide clear evidence that rice genotypes overexpressing PSY genes reduce CH₄ emissions, controlled environments do not fully replicate the complexity of field conditions, where factors such as temperature fluctuations and seasonal dynamics may play major roles. We consider our study a critical first step in uncovering mechanistic links between specific plant genotypes, root exudate chemistry, and CH₄ cycling. To validate and extend these findings to agronomically relevant conditions, field trials incorporating multi-omics profiling — similar in scope to our current approach — are necessary. Such future research will be essential to determine the generalizability and robustness of PSY-mediated reduced CH₄ emissions across various soil types and growing conditions.

Specific comments:

3. Line 104-106. I don't think the phylogenetic tree displayed in Supplemental Figure 1 b illustrates the claim that the authors make, and I lack a description of how the phylogeny was done. Is only the conserved domain included in the analysis? As far as I can tell, this analysis is not mentioned in the materials and methods section. If the authors want to compare the conserved domain, maybe this could be better shown with a sequence alignment.

Response: We thank the reviewer for pointing out the lack of methodological detail related to the phylogenetic analysis and for the helpful suggestion to revise this figure. We agree that the original version of Supplementary Fig. 1b may not have been the most appropriate representation to support our claim regarding sequence conservation of the mature PSY peptide across plant species.

In response, we have replaced the phylogenetic tree with a multiple sequence alignment and a sequence logo representation of the 13-amino acid mature PSY peptide domain from both *Arabidopsis* and rice. This revised visualization more clearly illustrates the high degree of conservation in the active peptide sequence, particularly around the conserved DY motif, which supports our hypothesis of functional conservation across species.

To reflect these changes, we have revised the figure and the main text, and added a detailed description of the sequence alignment and logo generation to the Materials and Methods section.

Manuscript now reads as (lines 118-124):

To identify putative rice PSY genes, we surveyed the Nipponbare and Kitaake genomes using bioinformatic approaches as previously described^{25,27}, and identified nine candidate

PSY genes (Supplementary Fig. 1a). Sequence alignment of the mature peptide domain revealed high conservation between Arabidopsis (AtPSYs) and rice (OsPSYs) peptides (Supplementary Fig. 1b), suggesting that OsPSY peptides may regulate rice root growth similarly to AtPSY peptides^{21–23,25,28}.

Materials and Methods section now reads as (lines 575-585):

Sequence alignment and peptide conservation analysis

To assess the conservation of the PSY peptides across species, we extracted the mature peptide sequences from *Arabidopsis thaliana* and *Oryza sativa* PSY family members. The peptide sequences were defined as the 13-amino acid mature domain, including the conserved aspartate-tyrosine (DY) motif, following the strategy described previously²⁵. A sequence logo was generated using WebLogo web interface (<http://weblogo.threplusone.com>)⁷², which provides a graphical representation of the multiple sequence alignment, where each position's stack height indicates sequence conservation, and symbol heights within the stack reflect the relative frequency of individual amino acids at that position.

Supplementary Figure 1. (b) Sequence logo generated using aligned 13-amino acid mature peptide sequences, including the conserved DY motif of the mature PSY peptide domain across *Arabidopsis thaliana* and *Oryza sativa*. Amino acids are color-coded based on chemistry.

4. Line 116-117. The authors hypothesize that LRR-RLKs may interact with the applied synthetic OsPSY peptides in root tissues and that this could influence root growth. Please provide an explanation for why.

Response: We agree that additional clarification is needed to support our hypothesis that leucine-rich repeat receptor-like kinases (LRR-RLKs) in rice roots may mediate the response to synthetic OsPSY peptides. This hypothesis is based on recent findings in *Arabidopsis*, where three different LRR-RLKs from subfamily XI were shown to function as direct ligand-binding receptors for PSY peptides and to regulate developmental processes such as root growth (Ogawa-Ohnishi et al., 2022, 10.1126/science.abq5735).

LRR-RLKs are highly conserved across plant lineages. We have revised the manuscript text to clarify this point.

Manuscript now reads as (lines 129-136):

We hypothesize that the root growth effect observed after synthetic peptide treatment may result from the presence of leucine-rich repeat receptor-like kinases (LRR-RLKs) in root tissues that recognize OsPSY1. It has been recently shown in Arabidopsis, that PSY peptides directly bind to LRR-RLKs from subfamily XI to trigger a signaling cascade that regulates root development²². The conservation of LRR-RLKs across plant species suggests that functionally similar receptors likely exist in rice and may mediate comparable developmental responses.

5. Line 143-143. The authors detect lower lignin content in PSY1-17 lines compared with Kitaake but not in the other transgenic lines. Please provide an explanation for this discrepancy between the phenotypes of the transgenic lines.

Response: At the time of the initial submission, lignin content had only been measured in PSY1-17 seedlings, which showed a significant reduction compared to Kitaake. For this revision, we have extended the analysis to include PSY2-34 lines. These new results demonstrate that the roots from seedlings that overexpress OsPSY2 also exhibit significantly reduced lignin levels compared to Kitaake. This finding is consistent with our observations in root systems from adult rice plants, where both PSY1-17 and PSY2-34 lines show reduced lignin accumulation (Fig. 2j). We have updated Figure 1i to include the new PSY2-34 data and revised the manuscript text accordingly to reflect these results:

Manuscript now reads as (lines 162-164):

Additionally, the amount of lignin in PSY1-17 and PSY2-34 plants was significantly lower compared with Kitaake (Fig. 1i).

Figure 1. (i) Lignin content (4 replicates of 15 complete seedling root systems).

6. Line 166-188. The structure of the text is difficult to follow since the results for EXP2 are presented before EXP1. I think the authors should restructure the text.

Response: We thank the reviewer for this helpful comment. We have reorganized the text based on this suggestion and created a new figure that includes all the transgenic lines.

Manuscript now reads as (lines 180-224):

To assess the effect of PSY overexpression on CH₄ emissions, we cultivated three independent transgenic rice lines — PSY1-9, PSY1-17, and PSY2-34 — along with the control Kitaake plants in a greenhouse using soils collected from a rice paddy in Davis, CA. We also included bulk soil controls that were not planted. Throughout the experiments, all rice lines developed similarly to Kitaake, with heading observed at 55 days after being transferred to the greenhouse (Fig. 2a). By 70 days post-transplant, both PSY1 and PSY2 genotypes were slightly shorter than Kitaake (Fig. 2b and Supplementary Fig. 2a), but they produced significantly more tillers (Fig. 2c and Supplementary Fig. 2b), resulting in comparable dry weights of stems and leaves per plant at the end of the plant life cycle (Fig. 2d). PSY1 genotypes showed partially filled panicles, fewer seeds, and lower thousand-grain weight (TGW) per plant, compared with Kitaake (Fig. 2e-g and Supplementary Fig. 2c). These phenotypes could be attributed to abnormal panicle development, as we found that the tips of the panicles in PSY1 genotypes often contained smaller and unfilled seeds, a phenotype not seen in PSY2 plants (Fig. 2h). Consistently, no significant differences in seed production occurred in the PSY2 genotypes compared with Kitaake (Fig. 2e-h).

We also investigated whether the longer seminal root phenotype observed in young seedlings persisted in older PSY plants. We found that PSY1-17 plants developed longer roots 30 days after transplanting, although their root dry biomass was significantly lower than that of Kitaake plants (Fig. 2k and Supplementary Fig. 2d-e). Similar differences in root dry biomass were observed at the end of the plant life cycle (Fig. 2i). Additionally, lignin content in the roots of adult plants revealed that, consistent with findings in young seedlings, PSY1-17 plants accumulated less lignin than Kitaake (Fig. 2j). Although PSY2 plants also had reduced lignin content, their root biomass remained comparable to that of Kitaake (Fig. 2i-j).

Despite these phenotypic differences, no significant differences in soil CH₄ emissions were observed between the PSY1-9, PSY1-17, and PSY2-34 genotypes (Fig. 2l). For these analyses, CH₄ emissions were measured *in situ* using a Picarro gas analyzer, and soil samples were collected for microbiological analyses throughout the rice growth period (i.e., 15, 30, 55, and 70 days after transplanting). Negligible CH₄ emissions were observed from the bulk soil controls, while differentiated emissions were recorded from pots with

PSY and Kitaake rice plants over the 10-week growth period (Fig. 2I). All PSY rice genotypes emitted substantially less CH₄ compared with Kitaake, with reductions of up to ~30% observed after heading (55 and 70 days; Fig. 2I). Notably, cumulative CH₄ emissions throughout the growth period for PSY1 and PSY2 genotypes showed an average decrease of ~38% and ~58% relative to Kitaake, respectively (Supplementary Fig. 3). Given the crucial role of microorganisms in the production and consumption of CH₄, we hypothesized that distinct microbial communities might accumulate in the rhizospheres of the different rice genotypes, potentially affecting the rice CH₄ emissions. Because both PSY1 and PSY2 genotypes displayed comparable reductions in CH₄ emissions, we focused on PSY1 genotypes for further study.

New **Figure 2I** and **Supplementary Figure 3**.

Fig. 2 Kitaake plants with elevated expression of *OsPSY1* or *OsPSY2* exhibited reduced CH₄ emissions. (I) Methane emission rates from different rice genotypes and bulk soil controls are normalized to the mean value obtained in Kitaake plants.

Supplementary Fig. 3 Methane emission rates over days after transplanting different rice genotypes. (b) Cumulative methane flux was calculated as the area under the curve

in (a) expressed in arbitrary units (A.U.). Different letters indicate significant differences, as determined by one-way ANOVA followed by Tukey's multiple comparison test ($P < 0.05$).

7. Line 212-219. The style of writing in this section resembles a Materials and Methods section. Please reformulate.

Response: We have moved some information to the Methods section and rephrased the sentences.

Manuscript now reads as (lines 227-234):

To explore the difference in root microbiomes, we extracted and sequenced total DNA and RNA from 72 soil samples (3 replicates for each of the bulk soils and 5 replicates for each planted with the PSY1 rice genotypes and Kitaake control, at each collection time point) (Supplementary Table 1). We used assembled ribosomal protein S3 (rpS3) sequences to perform an initial census of microbial diversity, including organism type and abundance³², in the 16 representative samples obtained by combining replicates. In total, 4,087 rpS3 sequences were identified and grouped into 1,656 non-redundant, approximately species-level clusters.

8. Line 220. The authors base their taxonomy on assembled rpS3 sequences. It has recently been shown that metagenome assemblies frequently fail to detect ribosomal sequences (Mise, K., Iwasaki, W. Unexpected absence of ribosomal protein genes from metagenome-assembled genomes (ISME COMMUN. 2, 118 (2022). <https://doi.org/10.1038/s43705-022-00204-6>). The authors should revise their taxonomic analysis accordingly and include more sequences as a base 0 for their taxonomic classification.

Response: The cited paper reports that ribosomal protein genes (including *rpS3*) can be undetected in metagenome-assembled genomes (not assemblies, as the reviewer notes), due to the dependence on scaffold tetranucleotide frequencies of metagenomic binning. We tested this expectation by profiling the metagenome-derived bins from our study for single-copy genes in two categories: (i) single-copy genes that are not ribosomal proteins, e.g., tRNA synthetases, and (ii) ribosomal protein genes. We asked whether or not a bin contained the selected marker genes (yes or no). Using bins ranging from near-complete to half-complete (approximately) as well as using bins that were at least ~78% complete, the detection of genes in these categories was not significantly different. More relevant to the concern as it relates to this paper, we note that our study analyzed all the *rpS3* sequences in scaffolds to profile the community structure (i.e., we did not rely on binned *rpS3* genes only), which should not have been affected by the above algorithmic flaw.

Our approach follows results of a previous study, which showed that the number of *rpS3* genes that could be recovered is > 5 times more than the number of genomes that could be binned in complex environmental samples, including soils (please see Fig. 3c in: Olm et al., Consistent metagenome-derived metrics verify and delineate bacterial species boundaries. *mSystems* 5.1 (2020). 10.1128/msystems.00731-19).

9. Line 232-234. The text in the manuscript does not match the text in the legend of Figure 3d:

Text in the manuscript: ...cultivation time has a large and statistically significant effect on the community structure (dissimilarity score of up to 0.539) but rice genotype does not (Fig. 3d)

Text in the Figure legend: ...for community structure (d), $R=0.539$, $P=0.001$ between rice genotypes (highlighted by ovals), and $R=0.0729$, $P=0.238$ between cultivation time;

Response: We apologize for the omission. The legend of Fig. 3d has been corrected.

The figure legend now reads as:

For community structure (d), $R=0.539$, $P=0.001$ between cultivation time (highlighted by ovals), and $R=0.0729$, $P=0.238$ between rice genotypes.

10. Line 287-290. This is a strong statement since the authors base their conclusion on transcriptome data, not functional microbiome studies. The authors should acknowledge this fact.

Response: We agree that our conclusion regarding hydrogen dynamics is based on transcriptomic data, but we do not know exactly what a “functional microbiome study” refers to. We have toned down the statement and clearly presented it as a hypothesis.

Manuscript now reads as (lines 302-306):

Based on the transcriptomic profile analysis, we hypothesize that the microbiomes in the PSY1 genotypes may produce less and consume more H_2 than those in Kitaake. As a result, the pool of hydrogen available for CH_4 production in soils from the PSY genotypes would be expected to be lower than that in the Kitaake control.

11. Line 301. Root secretion is an end product of root and soil microbial activity. The authors use axenic rice roots in their metabolic analyses, which may differ from real-life rhizospheric conditions. The authors should acknowledge this in the text.

Response: We deliberately chose axenic hydroponic conditions to collect early-stage root exudates, as this approach offers several experimental advantages: precise control over the chemical environment, sufficient recovery of fresh exudates, reduced mineral sorption artifacts, and minimal microbial transformation of metabolites (Zhalnina et al., 2018, 10.1038/s41564-018-0129-3). We acknowledge, however, that hydroponically derived exudation profiles may differ from those in soil-grown systems, where root–microbe interactions and soil matrix can alter both exudation rate and composition. Therefore, while the axenic method effectively isolates plant-derived metabolites, it fails to capture the dynamic interplay between plant roots and the soil microbiota. We have updated the Discussion to clarify this trade-off and to highlight the importance of combining hydroponic and soil-based exudate collection throughout plant development to uncover further ecological relevance.

Manuscript now reads as (lines 526-537):

It has been demonstrated that exudation rate and composition vary throughout plant development⁶⁸. In this study, we collected exudates from seedlings grown under axenic hydroponic conditions. This method offers several advantages: precise control over the chemical milieu, adequate recovery of fresh exudates, avoidance of mineral sorption biases, and minimization of microbial degradation⁶⁹. However, hydroponic growth can alter exudate profiles compared with soil systems, often increasing total carbon release but reducing metabolite diversity due to the absence of rhizosphere feedbacks⁷⁰. Given these methodological trade-offs, future studies could integrate both axenic hydroponic and soil-based exudate sampling across multiple developmental stages to gain a more comprehensive and ecologically relevant understanding of how PSY influences plant–microbe interactions and biogeochemical cycles.

12. Line 304-310. I think the authors could help the reader here by rearranging either the text in the manuscript or the figures. The authors write “root tissues or exudates” in the text and refer to Fig S5. a and e. However, Fig S5 a shows data for root exudates, and Fig S5 e shows data for root tissues. The remaining text in this paragraph follows a similar pattern. This makes the text unnecessarily difficult to follow.

Response: We agree that the current description could be clearer, especially in distinguishing results from root tissues versus exudates. To improve clarity, we have reorganized the text, aligning each description with the corresponding panels in Supplementary Figures 6-9 (originally Supplementary Figures 5-8). Additionally, we have updated the figure legends.

Manuscript now reads as (lines 316-349):

To explore the influence of rice phenotypes on rhizosphere microbiome activity, we performed metabolomic analysis on root tissues and exudates from axenic rice seedling roots to identify compounds secreted by the PSY1 genotypes. Metabolites were extracted using methanol and profiled on an HPLC-MS platform (Supplementary Dataset 1). Using gradient-based hydrophilic interaction liquid chromatography (HILIC) in positive-ion mode, we identified a total of 3,612 features across all exudate samples, of which 2,047 were shared between the Kitaake and PSY1 genotypes (Supplementary Fig. 6a). Principal component analysis (PCA) showed a clear separation between PSY1 and Kitaake exudate samples (Supplementary Fig. 6b and Supplementary Dataset 2). Among the shared features, 168 (8.3%) were significantly differentially abundant in PSY1 exudates ($|\log_2FC| > 0.5$, $p < 0.05$), including 146 novel and 15 depleted features (Supplementary Fig. 6c). A subset of these differentially abundant features was putatively identified and classified using Global Natural Products Social Molecular Networking (GNPS) (Supplementary Tables 3-5). Most of the differentially abundant metabolites in PSY1 root exudate samples belong to the same compound classes, including 'amino acids', 'alkaloids', and 'shikimates and phenylpropanoids' (Supplementary Fig. 6d). Similarly, in root tissues, we identified 2,237 shared features, with PCA indicating an even stronger genotypic separation (Supplementary Fig. 6e-f). Of these, 272 (12%) were differentially abundant in PSY1 tissues, featuring 208 new and 80 depleted metabolite signals (Supplementary Fig. 6g), again reflecting shifts in similar compound classes (Supplementary Fig. 6h). Overall, more features increased than decreased in PSY1 samples. Similar results were obtained from HILIC in negative ionization mode and reverse-phase (C18) in both positive and negative ionization modes and are detailed in Supplementary Dataset 2, Supplementary Tables 3-5, and Supplementary Figs. 7-9. Altogether, our analysis indicated significant changes in the abundance of compounds in the PSY1 root exudates compared with Kitaake. Approximately 90% of the metabolites enriched in the PSY1 metabolome are predicted to be involved in microbial gluconeogenic pathways (Supplementary Table 6). We also noted the presence of two putative flavonoids, isorhamnetin 3-rutinoside and malvidin-3-glucoside, that differentially accumulated in exudate samples from the PSY1 genotypes but were absent from Kitaake, a feature observed in previous studies in *Arabidopsis*²³.

Legends of **Supplementary Figures 6–9** read as:

Supplementary Figure 6. Metabolomic profiling of root exudates and tissues using HILIC (positive ionization mode). (a) Venn diagram of features detected in axenic root exudates from Kitaake and PSY1 seedlings. (b) Principal Component Analysis (PCA) of exudate features. (c) Volcano plot of differentially abundant metabolites in exudates (yellow = increased, blue = decreased; thresholds: $|\log_2FC| > 0.5$, $p < 0.05$). (d) Chemical classification of differentially abundant exudate metabolites, showing counts of increased/new vs. decreased/depleted compounds. (e–h) Analogous analyses for root

tissue samples: **(e)** Venn diagram, **(f)** PCA, **(g)** volcano plot, and **(h)** chemical classification. Numbers indicate features unique to or shared between genotypes and how many were significantly up- or down-regulated.

Supplementary Figure 7. Metabolomic profiling of root exudates and tissues using HILIC (negative ionization mode). **(a)** Venn diagram of features detected in axenic root exudates from Kitaake and PSY1 seedlings. **(b)** Principal Component Analysis (PCA) of exudate features. **(c)** Volcano plot of differentially abundant metabolites in exudates (yellow = increased, blue = decreased; thresholds: $|\log_2 FC| > 0.5$, $p < 0.05$). **(d)** Chemical classification of differentially abundant exudate metabolites, showing counts of increased/new vs. decreased/depleted compounds. **(e–h)** Analogous analyses for root tissue samples: **(e)** Venn diagram, **(f)** PCA, **(g)** volcano plot, and **(h)** chemical classification. Numbers indicate features unique to or shared between genotypes and how many were significantly up- or down-regulated.

Supplementary Figure 8. Metabolomic profiling of root exudates and tissues using C18 (positive ionization mode). **(a)** Venn diagram of features detected in axenic root exudates from Kitaake and PSY1 seedlings. **(b)** Principal Component Analysis (PCA) of exudate features. **(c)** Volcano plot of differentially abundant metabolites in exudates (yellow = increased, blue = decreased; thresholds: $|\log_2 FC| > 0.5$, $p < 0.05$). **(d)** Chemical classification of differentially abundant exudate metabolites, showing counts of increased/new vs. decreased/depleted compounds. **(e–h)** Analogous analyses for root tissue samples: **(e)** Venn diagram, **(f)** PCA, **(g)** volcano plot, and **(h)** chemical classification. Numbers indicate features unique to or shared between genotypes and how many were significantly up- or down-regulated.

Supplementary Figure 9. Metabolomic profiling of root exudates and tissues using C18 (negative ionization mode). **(a)** Venn diagram of features detected in axenic root exudates from Kitaake and PSY1 seedlings. **(b)** Principal Component Analysis (PCA) of exudate features. **(c)** Volcano plot of differentially abundant metabolites in exudates (yellow = increased, blue = decreased; thresholds: $|\log_2 FC| > 0.5$, $p < 0.05$). **(d)** Chemical classification of differentially abundant exudate metabolites, showing counts of increased/new vs. decreased/depleted compounds. **(e–h)** Analogous analyses for root tissue samples: **(e)** Venn diagram, **(f)** PCA, **(g)** volcano plot, and **(h)** chemical classification. Numbers indicate features unique to or shared between genotypes and how many were significantly up- or down-regulated.

13. Line 355-378. The metabolic models need further clarifications:
a. The authors use PSY over secreted compounds in their models. How do the models look if the corresponding over-secreted compounds from the wild-type comparator were used?

Response: We thank the reviewer for the suggestion. The only identified compounds over-secreted by the wild-type Kitaake are two glycerolipids, TG(18:1/18:2/18:2) and TG(16:0/16:0/18:3), and one dipeptide containing leucine. We input the representative of glycerolipids, triglyceride (KEGG: C00422), and the leucyl-leucine (KEGG: C11332) into our metabolic models and found that none of the organisms could metabolize these compounds. Consequently, the Kitaake over-secreted metabolites did not affect the predicted H₂ flux. We have revised the text accordingly.

Manuscript now reads as (lines 387-399):

...predicted H₂ fluxes in response to variations in PSY1-secreted exudate compounds and O₂ concentrations inferred from aerenchyma densities of the different rice genotypes (Supplementary Table 8). The only identified compounds under-secreted by PSY1 compared with Kitaake were two glycerolipids, putatively 1-oleoyl-2,3-di-linoleoyl-sn-glycerol and 1,2-di-hexadecanoyl-3-(9Z,12Z,15Z-octadecatrienoyl)-sn-glycerol, and one dipeptide containing leucine (Supplementary Table 3). However, none of the modeled H₂-related organisms can metabolize these compounds. As a result, the PSY1 under-secreted metabolites are predicted to have no effect on H₂ fluxes. In contrast, when over-secreted compounds increase from 0.01 to 0.05 fold, all three modeled H₂-producing bacteria from different phyla (Bacteroidia, Ignavibacteriota, and Moranbacteria) show the same trend, with maximal produced H₂ decreasing (Fig. 7a-c).

b. Also, does the total amount of secreted compounds differ between the genotypes or is only the composition. If there is an increased total amount of compounds released, this will likely have a large effect on methane emissions.

Response: To determine whether the total amount of secreted compounds differed between genotypes, we quantified the concentration of non-purgeable organic carbon (NPOC) in the exudate samples used for metabolomic analysis. The results showed no significant differences in total carbon content among PSY genotypes and Kitaake exudates, indicating that the primary variation lies in compound composition rather than total exudation rate. We have added this clarification to the end of the metabolomics section in the revised manuscript.

Manuscript now reads as (lines 350-355):

To assess whether these metabolic differences reflected changes in the total amount of carbon released, we measured the concentration of non-purgeable organic carbon (NPOC) in the root exudate samples. No significant differences in total carbon content were detected between Kitaake and PSY genotypes, indicating that PSY overexpression primarily alters the composition, instead of the overall quantity, of secreted metabolites (Supplementary Fig. 10).

Supplementary Figure 10. Total organic carbon content in root exudates. Non-purgeable organic carbon (NPOC, mg L^{-1}) was measured in exudates collected from 8-day-old rice seedlings of Kitaake, PSY1, and PSY2 genotypes, along with a water-only control. Bars represent the mean \pm SEM from eight biological replicates per genotype. Different letters indicate significant differences, as determined by one-way ANOVA followed by Tukey's multiple comparison test ($P < 0.05$).

Materials and Methods now read as (lines 939-954):

Non-purgeable organic carbon (NPOC) measurement

Frozen root exudates collected as described previously were thawed at 4°C and then diluted to 8 mL using Milli-Q water. Samples were analyzed on a Shimadzu TOC-L CPH total organic carbon analyzer equipped with TNM-L and ASI-L modules using UHP synthetic air mix as carrier gas. Calibration curves for non-purgeable organic carbon were prepared using potassium hydrogen phthalate. Standards and samples were analyzed under the same settings; blanks and control samples were included for quality control purposes. Sample measurements were set up with 1 wash step, followed by automated acid addition at 1.5% 1M HCl with a 2.5 minute sparging period at 80 mL/min to remove inorganic carbon and purgeable carbon. Acidified, sparged samples containing non-purgeable organic carbon were then injected at 50 μL for analysis using a max 4.5 min integration window for each carbon measurement; this was performed with 3-5 technical replicate injections with the max cv set at 2%. Measurements were corrected for dilution factors and then reported as ppm C (mg of C per L of exudate).

c. The authors identify 17 microbial genomes with increased transcriptional activities in soils of PSY genotypes of which five had H₂-consuming hydrogenases, please provide information on the taxonomy of these microbial genomes.

Response: This information was initially included in the Supplementary Table 7 only. To make it clear, we have now added it to the main text.

Manuscript now reads as (lines 370-374):

The genomes of five of the 17 encode H₂-consuming hydrogenases, including those affiliated with Pyrinomonadaceae, Burkholderiaceae, Haliangiaceae, Rhodocyclaceae, and Dissulfurspiraceae, but none contain H₂-producing enzymes (Supplementary Table 7).

d. The authors then build the models on three of these microbial genomes. How were these three genomes selected? Which genomes of the 17 were used, and what were the species? All this information is needed to evaluate the models.

Response: First, we used metabolic modeling to investigate how rice physiological features, particularly the over-secreted compounds, affect H₂-related organisms and the associated H₂-cycling process. Thus, only the genomes that have H₂-producing or H₂-consuming hydrogenases were selected. Second, metabolic modeling requires high-quality genomes. Here, we filtered those of predicted completeness > 90% and contamination < 5%. Finally, we chose genomes from different phyla to better represent the H₂-related population. Based on these criteria, H₂-producing genomes affiliated with Bacteroidia, Ignavibacteriota, Moranbacteria, and H₂-consuming genomes from Acidobacteriota, Proteobacteria (one of the 17 up-regulated genomes), and Deinococcota were ultimately selected for metabolic modeling. We have added this and the taxonomic information of the selected genomes to the manuscript.

Manuscript now reads as (lines 407-411):

The H₂-consuming organisms from Acidobacteriota, Proteobacteria, and Deinococcota are modeled to oxidize more H₂ (and accordingly yield more biomass) in the presence of higher O₂ concentrations that are associated with higher aerenchyma formation in PSY1 rice genotypes (Fig. 7d-f and Supplementary Fig. 13d-f).

In **Figure 7** legend:

Genome name abbreviations: Bacteroidia_37, REG_22_10_11_KiT_metabat2_37 in Supplementary Table 7; Ignavibacteriota_4780, REG_22_11_18_KiT_metabat2_4780; Moranbacteria_75, REG_22_11_18_KiT_metabat2_75; Acidobacteriota_95, REG_22_09_28_Psy117_metabat2_95; Proteobacteria_149,

REG_22_10_11_Psy117_concoct_149; Deinococcota_10534, REG_22_09_28_KiT_vamb_10534. All detailed information of these genomes is present in Supplementary Table 7.

In **Methods** (lines 976-983):

We selected microbial genomes for metabolic modeling based on the following criteria: (1) genomes contain H₂-producing or H₂-consuming hydrogenases; (2) genomes are predicted to have completeness > 90% and contamination < 5%; (3) genomes are affiliated with different phyla. Three H₂-producing genomes from Bacteroidia, Ignavibacteriota, and Moranbacteria, and three H₂-consuming genomes from Acidobacteriota, Proteobacteria, and Deinococcota were finally selected and annotated using the App “Annotate Genome/Assembly with RASTtk” (v1.073) in KBase¹²⁰.

In **Data availability** (lines 1018-1019):

Microbial metabolic models have been deposited in figshare (<https://doi.org/10.6084/m9.figshare.29115692>).

e. Line 375-378. The final sentence should be clarified since it conveys conflicting results.

Response: The PSY rice genotypes have more aerenchyma and a distinct exudate profile compared with the Kitaake control. Our modeling suggests that the resulting higher O₂ concentrations and PSY1 exudates promote H₂ oxidation and biomass yield of H₂-consuming organisms. We have rephrased the sentence for clarity.

Manuscript now reads as (lines 407-416):

The H₂-consuming organisms from Acidobacteriota, Proteobacteria, and Deinococcota are modeled to oxidize more H₂ (and accordingly yield more biomass) in the presence of higher O₂ concentrations that are associated with higher aerenchyma formation in PSY1 rice genotypes (Fig. 7d-f and Supplementary Fig. 13d-f). The PSY1 over-secreted exudate compounds have a negligible influence on H₂ oxidation, yet still contribute substantially to accumulation of biomass of H₂-consuming organisms (Supplementary Fig. 13d-e). These modeling results support the metagenomics-based hypothesis that PSY1 rice genotypes enhance the growth and activity of H₂-consuming organisms due to their distinct aerenchyma characteristics and exudate profiles.

f. Finally, if the species used in the models can be cultivated, it should also be possible to verify the models experimentally.

Response: We thank the reviewer for the suggestion. We have acknowledged this limitation in the Discussion and would like to try isolation and cultivation in future experiments.

Manuscript now reads as (lines 518-525):

Overall, our metabolic modeling predicts that the exudate profiles of PSY1 rice genotypes may inhibit the activities of H₂-producing microorganisms and stimulate those of the H₂-consuming counterparts. This result suggests that in the PSY1 rhizosphere, the bacterial metabolic activities further diminish the pool of H₂ available for methanogenesis thus mitigating overall CH₄ emissions (Fig. 8). Future studies could isolate and co-cultivate the organisms used in the models to validate the inferred microbial responses to root-derived environmental changes.

14. Line 398-404. The authors state that methane cannot be derived from acetate as low activity of known aceticlastic methanogens was found, and they suggest that acetate might not even be produced or only be present at low levels. Acetate, together with H₂, is the final products from degradation of all organic compounds in an anaerobic environment, and thus, it seems very unlikely that no acetate is produced. Moreover, the authors do not mention the possibility of the alternative route for methane production from acetate, that is syntrophic acetate oxidation (SAO). This route for methane production has been found to be present in rice soil and is likely to be one pathway for hydrogen production feeding hydrogenotrophic methanogens. The text has to be revised accordingly. Moreover, the activity of this group would also be relevant to explore in the transcriptomic data, e.g. the WL pathway, possibly combined with the glycine cleavage system recently shown important for SAO. This pathway is also relevant for H₂ oxidizing acetogens, possibly representing an important H₂ oxidizing bacterial group, in addition to sulfate, iron and reducing.

Response: We agree with the reviewer that there must be acetate production. Based on the literature, the substrate affinity of acetate of methanogens appears to be much higher than that of hydrogen (0.2-1.2 mM acetate versus 2.8-10 Pa H₂). Meanwhile, other co-existing organisms may outcompete methanogens for the in situ produced acetate, e.g., for the reduction of nitrate and sulfate. Thus, we hypothesize that the hydrogenotrophic pathway was selected as the dominant methanogenesis pathway under such environmental conditions, which is consistent with previous observations (e.g., 10.1038/s41558-023-01872-5, 10.3389/fmicb.2012.00004).

We also thank the reviewer for mentioning the possibility of syntrophic acetate oxidation (SAO) for methane production. SAO bacteria use the reverse Wood-Ljungdahl pathway (sometimes combined with the glycine cleavage system) and hydrogenases to oxidize acetate to CO₂ and H₂, the substrates for hydrogenotrophic methanogenesis. Thus, we

investigated the key enzymes in our 36 H₂-producing microbial genomes. However, none of them encode complete Wood-Ljungdahl pathways, although some genomes are predicted to be > 90% complete (by CheckM2), precluding their ability to perform SAO. This is not unexpected because most of our H₂-producing genomes are from the phyla Bacteroidetes (GTDB taxon: Bacteroidota) and Chloroflexi (GTDB taxon: Chloroflexota) (Supplementary Table 7). On the contrary, most of the reported SAO bacteria that co-exist with methanogens belong to the phylum Firmicutes (GTDB taxon: Bacillota). Bacteria of this phylum were rarely detected in our study (< 1%; Fig. 3). Of the three organisms detected, one affiliates with the family Symbiobacteriaceae and two with the class Negativicutes (Supplementary Table 7), and none encode the Wood-Ljungdahl pathway.

We also analyzed our 55 H₂-consuming microbial genomes and found no H₂-oxidizing homoacetogenesis pathways. This is not surprising because none of these genomes are from the class Clostridia of Firmicutes, which includes most of the discovered homoacetogens.

Overall, we acknowledge the possibility of syntrophic acetate oxidation for methane production and the potential of homoacetogenesis for H₂ competition, but they appeared to make a minor contribution. We have added the above information and discussion in the related text.

Manuscript now reads as (lines 432-455):

Previous rice field studies using isotope-labeled substrates revealed that the presence of electron acceptors such as sulfate and nitrate channeled acetate towards CO₂, and that CH₄ was mainly produced from H₂/CO₂-dependent methanogenesis^{51,52}. Dependence on H₂ is likely explained by acetate concentrations that are insufficient for methane production. Acetoclastic methanogens (e.g., *Methanosarcina*) require 0.2-1.2 mM acetate to activate CH₄ formation^{53,54} whereas hydrogenotrophic methanogens (e.g., *Methanobacterium*) only need 2.8-10 Pa H₂^{55,56}. Low acetate concentrations may be due to the rapid *in situ* utilization by co-existing organisms that outcompete methanogens for the acetate. One possible route for acetate consumption is syntrophic acetate oxidation (SAO) that uses the reverse Wood-Ljungdahl (WL) pathway (sometimes combined with the glycine cleavage system) and hydrogenases to produce CO₂ and H₂, the substrates for hydrogenotrophic methanogenesis⁵⁷⁻⁵⁹. However, none of our hydrogenase-harboring microbial genomes encode complete WL pathways, although some unclassified Firmicutes members that may include SAO bacteria were detected at low abundance (< 1%; Fig. 3). An alternative way for acetate consumption is the reduction of nitrate and sulfate, observed in previous studies^{51,52}. This activity is indicated by the expression of nitrate and sulfate reductases in our system (Supplementary Figs. 14 and 15). The

substrates (i.e., ~2.0 mM of both nitrate and sulfate) were periodically added to water used to fertilize the rice plants during growth (Supplementary Table 2). Given the absence of WL pathways in hydrogenase-containing microorganisms, the homoacetogenesis process that relies on H₂ oxidation to produce acetate appears to be negligible in our study.

Reviewer #2 (Remarks to the Author):

Response: We appreciate the reviewer and the journal for the huge support for early-career researchers.

Reviewer #3 (Remarks to the Author):

Overall comments

1. The manuscript is well-structured and provides a comprehensive analysis demonstrating the methane emission mitigation effect of PSY in rice.

Response: We thank the reviewer for their time and overall positive comment on the manuscript.

However, the metagenomic and metabolomic data interpretations should provide more concrete evidence of causality rather than correlation. For example, while PSY-expressed plants show altered exudates, how do we confirm that these exudates are indeed directly responsible for microbial shifts?

Response: We thank the reviewer for highlighting the importance of distinguishing causality from correlation in linking PSY-altered exudates to microbial activity. To address this, we conducted additional soil incubation experiments using rice root exudates. These exudates were added to paddy soil collected from the same field site used in our greenhouse experiments, and then CH₄ emissions were measured over time. We found that soil treated with Kitaake exudates consistently exhibited higher CH₄ emissions than soil treated with PSY1 or PSY2 exudates, recapitulating the patterns we observed in planta. Importantly, soil incubated with water alone did not emit measurable CH₄, confirming that microbial activity was driven by the chemical composition of the exudates. This response is consistent with prior studies showing that root exudates directly stimulate CH₄ production in paddy soils, for example, Aulakh et al., 2001 (<https://doi.org/10.1055/s-2001-12905>).

These new results provide functional evidence that PSY-driven changes in exudate chemistry directly influence microbial activity in the rhizosphere, independent of other plant phenotypes. We have added these results to the revised manuscript. Please also see our response to Comment 1 from Reviewer #1 (above).

2. The microbial community analysis (Figure 3) could benefit from clearer labeling—perhaps grouping similar microbial taxa together visually.

Response: We thank the reviewer for the suggestion. The rpS3 protein sequences in Fig. 3a were clustered at 99% identity before the alignment and the phylogeny reconstruction. We have now modified the figure by collapsing the clades that are from the same phylum for clearer labeling. We copied the updated Fig. 3 below for your convenience.

Fig. 3 Microbial community structure and activity.

3. The authors propose that PSY exudates selectively enhance H_2 -oxidizing microbes while suppressing H_2 -producing ones. This is intriguing, but it is unclear whether:

-This effect is directly caused by specific exudate compounds or just root-associated environmental changes.

Response: We thank the reviewer for this thoughtful question. To begin addressing whether the microbial shifts we observed are directly caused by PSY-modulated exudate composition or by broader root-associated environmental changes, we conducted additional soil incubation experiments using exudates collected from PSY-overexpressing genotypes and Kitaake seedlings, as described above. By applying these exudates to agronomic soil under controlled conditions, we found that soils treated with Kitaake-derived exudates consistently emitted significantly more CH_4 than those treated with exudates from PSY-overexpressing plants. These results closely mirror the trends observed in our greenhouse experiments, providing evidence that root exudate composition alone can influence microbial activity related to methane cycling.

While our current study isolates the effect of exudates, we cannot disentangle whether it was caused by a specific compound or by a combination of multiple compounds. We also

cannot fully preclude the possibility that root-associated environmental changes may contribute to microbial shifts (e.g., by altering local redox conditions). Due to the scale and complexity of these experiments, these additional layers of mechanistic insight will be pursued as part of a follow-up project. We have discussed this in the revised manuscript.

Manuscript now read as (lines 501-507):

While our soil exudate incubation experiments clearly show the effect of rice root exudates on microbial shifts and associated CH₄ emissions, it is unclear whether this is caused by a specific exudate compound or by a combination of multiple compounds. It is also possible that root-associated environmental changes (e.g., altered redox conditions) may affect the rhizosphere microbiome. Future research is warranted to investigate these possibilities.

-it is unclear whether the microbial shifts persist beyond controlled conditions.

Response: We acknowledge that validation under field settings would be essential to confirm the generalizability of our greenhouse findings. However, conducting a field trial utilizing transgenic rice genotypes, as well as comprehensive transcriptome, metabolome, and microbiome profiling, would require extensive time and regulatory approval. We anticipate this effort would span at least 18 months, due to planting schedules, seasonal sampling, and post-harvest analyses. We have added a statement to the Discussion to acknowledge this limitation and to describe our plans for upcoming field-based validation studies. Please see the response to Comment 2 from Reviewer #1 in the preceding.

-it is unclear whether the observed metabolic effects (e.g., gluconeogenic acid enrichment) are due to PSY expression or indirect plant growth effects.

Response: We appreciate the reviewer's insight regarding whether the observed changes in exudate chemistry are a direct result of PSY signaling cascade or indirectly driven by alterations to root structure, including longer seminal roots, increased aerenchyma, and reduced lignin. While elucidating the mechanistic pathway underlying the changes in exudate chemistry would be a significant contribution, it would require additional experiments, and we believe these experiments fall beyond the scope of this manuscript.

It is worth noting that recent work has demonstrated that developmental changes in endodermal integrity, such as delayed Casparian strip formation, can lead to amino acid leakage and altered spatial colonization by the root microbiota (Tsai et al., 2025, 10.1126/science.adu4235). In our system, PSY-overexpressing lines exhibit reduced lignin deposition, and it would be interesting to test whether this correlates with a delay in

Casparian strip formation, which may similarly compromise barrier function and contribute to the observed enrichment of gluconeogenic compounds in the rhizosphere.

4. What are the agronomic trade-offs of PSY-expressing rice? The paper mentions shorter stature, reduced grain weight, and other agronomic traits. How might these factors impact yield? Similarly, Iqbal et al. (2021) demonstrated that while methane emissions were significantly reduced, there was a trade-off with rice yield. Both OsLSD (Iqbal et al.) and OsPSY influence root biomass and aerenchyma development, which typically leads to reduced rice growth..

Response: As shown in Figure 2, PSY1 genotypes exhibit reduced plant height and altered panicle development, which are associated with a decrease in thousand-grain weight (TGW). In contrast, PSY2 lines maintain both yield and shoot biomass comparable to Kitaake. Notably, although PSY2 genotypes do not display yield penalties, they exhibit root phenotypes similar to PSY1 lines, including longer seminal roots, reduced lignin deposition, and increased aerenchyma formation. These observations suggest that while PSY1 and PSY2 belong to the same peptide family, their differential phenotypes may result from distinct affinities to specific LRR-RLK receptors, leading to divergent developmental outcomes.

Aerenchyma formation has been linked to changes in methane emissions. The OsLSD study referenced by the reviewer describes a rice mutant with decreased aerenchyma and reduced CH₄ production, but also reported yield penalties. In contrast, our results show that PSY2 lines exhibit increased aerenchyma and reduced CH₄ emissions without compromising grain yield. This indicates that, in our system, methane mitigation can be decoupled from agronomic penalties.

5. Is it differently expressed PSY on other parts? (panicle, leaf etc)

Response: Recent comprehensive analyses in *Arabidopsis thaliana* using reporter lines and transcriptome databases have revealed that PSY-family peptides and their cognate receptors exhibit distinct, organ-specific expression profiles (Ogawa-Ohnishi et al., 2022, 10.1126/science.abq5735). Although similar detailed expression profiling has yet to be performed in rice, we have leveraged available public transcriptomic data (Supplementary Fig. 1) to begin characterizing OsPSY expression patterns across various tissues. Additionally, the transgenic lines used in this study express *OsPSY1* and *OsPSY2* under a constitutive promoter, ensuring broad expression across roots, shoots, and panicles. This broad overexpression supports the potential for PSY peptides to interact with LRR-RLK receptors across multiple plant organs.

6. PSY decrease lignin contents, please provide correlation with lignin and methane emission.

Response: Suberin and lignin form key components of the outer apoplastic barrier in rice roots through deposition in the exodermis, which restricts radial oxygen loss (ROL), a trait vital for waterlogging tolerance that also controls the oxygen reaching the rhizosphere (Peralta Ogorek et al., 2021 & 2023, 10.1111/nph.17474, 10.1071/FP23133). Although this link remains hypothetical, the observed decrease in lignin content in PSY-overexpressing lines suggests a weakened root ROL barrier, which may enhance oxygen diffusion into the rhizosphere. This increased oxygen availability likely promotes aerobic methane oxidation, offering a plausible explanation for the reduced CH₄ emissions we recorded.

7. How do the haplotypes of the PSY gene appear in natural variants, and what are their potential effects?

Response: While investigating haplotype variation and natural diversity in the 10,000 sequenced rice lines is a valuable direction for future research, it falls outside the scope of the present study.

Detail comments

-p. 16 Line 1: It is not appropriate to compare the loss-of-function GS3 allele with PSY overexpression, as GS3 is a natural variant, whereas PSY is an overexpressed mutant. Therefore, the methane mitigation effect should be compared with other gene-edited varieties or genes.

Response: We agree that comparison with the loss-of-function GS3 allele may not be the most suitable comparison, given that we are using a transgenic approach. We have removed that phrase from the text to avoid confusion.

-(p. 5 Line : 90 Introduction) Please provide sufficient information about PSY gene and role of PSY1 and PSY2

Response: We thank the reviewer for highlighting the need to include more background information on PSY peptides and their specific roles. In response, we have expanded the relevant sections of the manuscript to provide additional context.

Manuscript now reads as (line 90-106):

Previously, we and others have shown that small tyrosine-sulfated peptides, such as the PLANT PEPTIDES CONTAINING SULFATED TYROSINE (PSY) family, play central roles in coordinating root development. Acting as extracellular signals, these peptide hormones synchronize cellular activities across tissues by binding to leucine-rich repeat receptor-like kinases (LRR-RLKs), thereby triggering downstream signaling cascades¹⁶. A critical post-translational modification, sulfation of a tyrosine residue by

TYROSYLPROTEIN SULFOTRANSFERASE (TPST), is essential for receptor recognition and signaling efficiency^{17–20}. In *Arabidopsis thaliana*, overexpression of PSY genes enhances root growth by promoting cell elongation and increasing the size of mature cortical cells^{21–23}. These peptides exert their effects through three known LRR-RLK receptors — PSYR1, PSYR2, and PSYR3 — recently identified as direct mediators of PSY signaling^{22,24}. Notably, PSY1 signaling in *Arabidopsis* activates genes involved in secondary metabolism that regulate the extent of cell elongation and transition to maturation. Some of these metabolites are subsequently secreted into the rhizosphere as part of the root exudate profile^{21,23,25}.

-(p. 23 Line : 654 Methods) Please provide how many bp read to analyze metagenomics analysis (rpS3)

Response: We used 150-bp paired reads for both metagenomic and metatranscriptomic sequencing. The rpS3 analysis was based on assembled scaffolds (not reads).

We have rephrased the sentences, which now read as (lines 744-745 and line 760):

Raw 150-bp paired reads from metagenomic sequencing were processed using BBDuk...

For metatranscriptomics, raw 150-bp paired reads were pre-treated using BBDuk...

Reviewer #1 (Remarks to the Author):

Review of Manuscript NCOMMS-24.785229

I thank the authors for their updated and improved manuscript. The authors have provided tangible answers to most of my comments and concerns regarding the original submission. I only have a few additional comments:

1. I note that the authors have chosen not to perform field trials with their transgenic plants. While such experiments would be informative, acknowledging this limitation suffices, given the quality of the data now presented in the manuscript.

Response: We thank the reviewer for recognizing the quality of the data presented in the manuscript. As mentioned in the previous round, this limitation has been acknowledged and discussed in the manuscript (lines 533-543).

2. The authors do not understand my reference to “functional microbiome studies” (comment 10 in the first round of revision). Functional microbiome studies would include cultivation of microbiome isolates and studies of H₂ production and CH₄ emissions under controlled conditions. As noted by the authors in lines 524-524, such experiments could be used to further test and substantiate the models presented in Figure 7. In this context, it is also important to acknowledge the limitations of transcriptome studies and that they are not the same as functional studies. However, I think the text edits the authors have now made clarify the limitations of their study sufficiently.

Response: As noted by the reviewers, the limitations of the study have been clearly clarified in the manuscript. We thank the reviewer for all the constructive comments.

Reviewer #2 (Remarks to the Author):

Response: We appreciate the reviewer and the journal for the huge support for early-career researchers.